DOI: 10.1038/s41467-018-07176-z | OPEN

# Human adipose glycerol flux is regulated by a pH gate in AQP10

Kamil Gotfryd [1], Andreia Filipa Mósca[2], Julie Winkel Missel[1], Sigurd Friis Truelsen [3], Kaituo Wang [1], Mariana Spulber[4], Simon Krabbe [5], Claus Hélix-Nielsen[3,4], Umberto Laforenza [6], Graça Soveral [2], Per Amstrup Pedersen [5] & Pontus Gourdon [1,7]

Obesity is a major threat to global health and metabolically associated with glycerol homeostasis. Here we demonstrate that in human adipocytes, the decreased pH observed during lipolysis (fat burning) correlates with increased glycerol release and stimulation of aquaglyceroporin AQP10. The crystal structure of human AQP10 determined at 2.3 Å resolution unveils the molecular basis for pH modulation—an exceptionally wide selectivity (ar/R) filter and a unique cytoplasmic gate. Structural and functional (in vitro and in vivo) analyses disclose a glycerol-specific pH-dependence and pinpoint pore-lining His80 as the pH-sensor. Molecular dynamics simulations indicate how gate opening is achieved. These findings unravel a unique type of aquaporin regulation important for controlling body fat mass. Thus, targeting the cytoplasmic gate to induce constitutive glycerol secretion may offer an attractive option for treating obesity and related complications.

[1] University of Copenhagen, Department of Biomedical Sciences, Nørre Allé 14, DK-2200 Copenhagen N, Denmark. [2] Universidade de Lisboa, Research Institute for Medicines (iMed.ULisboa), Faculty of Pharmacy, Av. Prof. Gama Pinto, 1649-003 Lisbon, Portugal. [3] Technical University of Denmark, Department of Environmental Engineering, Bygningstorvet Building 115, DK-2800 Kgs Lyngby, Denmark. [4] Aquaporin A/S, Nymøllevej 78, DK-2800 Lyngby, Denmark. [5] University of Copenhagen, Department of Biology, Universitetsparken 13, DK-2100 Copenhagen OE, Denmark. [6] University of Pavia, Department of Molecular Medicine, Human Physiology Unit, Via Forlanini 6, I-27100 Pavia, Italy. [7] Lund University, Department of Experimental Medical Science, Sölvegatan 19, SE-221 84 Lund, Sweden. These authors contributed equally: Andreia Filipa Mósca, Julie Winkel Missel, Sigurd Friis Truelsen. Correspondence and requests for materials should be addressed to P.A.P. (email: papedersen@bio.ku.dk) or to P.G. (email: pontus@sund.ku.dk)

Over the last decades the incidence of medical conditions related to obesity, such as type 2 diabetes and cardiovascular disease, has dramatically increased, reaching epidemic proportions[1,2]. Formation (lipogenesis) and breakdown (lipolysis) of lipids such as triacylglycerols (TAGs) in adipocytes, the main cell type of adipose tissue, are hallmarks of body fat homeostasis[3]. Lipolysis is a lipase and pH-dependent process[4–6] that alongside dietary supply delivers the majority of plasma free fatty acids and glycerol[7] required for fueling peripheral tissues[8,9]. Uptake and release of glycerol from the small intestine (duodenal enterocytes), adipocytes and other cell types, are primarily facilitated by a subclass of aquaporins (AQP), the water and glycerol-conducting aquaglyceroporins (AQP3, 7, 9 and 10; Fig. 1a and Supplementary Fig. 1)[9–11]. Furthermore, mice aquaglyceroporin AQP7 knockouts accumulate glycerol and TAGs, and develop enlarged adipocytes and obesity with age. Thus, glycerol and aquaglyceroporin-induced glycerol flux are central elements of fat accumulation and the pathophysiology of obesity[12,13]. Nevertheless, the molecular principles that regulate glycerol flow across cellular membranes in the body remain enigmatic. The interplay between lipolysis and glycerol flux is obscure, and human aquaglyceroporins are primarily believed to be controlled through trafficking (e.g., catecholamine/insulin-dependent subcellular re-organization of AQP7 in adipocytes)[10,14], as structural information is available only for homologs from lower organisms[15–17]. Here, we report the crystal structure of hAQP10, the only human aquaglyceroporin that becomes stimulated by pH reduction, in agreement with the altered cellular conditions observed during lipolysis in human adipocytes. In contrast to other known aquaporin structures, hAQP10 displays pH-dependent glycerol-specific gating at the intracellular interface, rather than flux governed by the classical selectivity filter. Specifically, we identify H80 as a critical part for this regulation, being the pH-sensor that propagates structural rearrangements leading to the opening of hAQP10, upon double protonation associated with low pH conditions.

## Results

**Low pH stimulates adipocyte glycerol flux through hAQP10.** To link the rate of glycerol release to changes in cytosolic pH in human adipocytes, we induced either lipogenesis (insulin supplementation) or lipolysis (isoproterenol supplementation)[9,13]. Whereas insulin treatment did not affect intracellular pH or glycerol export (Fig. 1b and Supplementary Fig. 2), induction of lipolysis resulted in internal acidification and stimulation of glycerol release. Only a few reports hint at a pH-dependence of mammalian aquaglyceroporin-mediated flux[9,18,19]. We therefore assessed the pH effect on water and glycerol flow across membrane vesicles prepared from human adipocytes challenged with osmotic gradients (Fig. 1b and Supplementary Fig. 3a). Whereas the permeability to water ($P_f$) was pH-insensitive, glycerol passage ($P_{gly}$) increased at low pH (pH 7.4 vs. 5.5). To identify the responsible protein(s), all four human aquaglyceroporins and water-strict (orthodox) hAQP2 serving as control were isolated as green fluorescent protein (GFP)-fusions (Supplementary Table 1). Following reconstitution into biomimetic vesicles (polymersomes)[20], we investigated the flow rates at the equivalent pH upon osmotic stress (Fig. 1c, Supplementary Fig. 3b and Supplementary Table 2). The water permeability was unchanged for all tested AQPs, suggesting pH-insensitive water diffusion. Glycerol conductance was in contrast highly pH-dependent. As expected, orthodox hAQP2$_{GFP}$ displayed no glycerol permeation. hAQP3$_{GFP}$, 7$_{GFP}$ and 9$_{GFP}$ were permeable to glycerol only at pH 7.4. Only hAQP10$_{GFP}$ allowed glycerol flux at pH 5.5, whereas the flow at pH 7.4 was markedly reduced, in agreement with the

adipocyte-based data. As an additional control, we detected plasma membrane-localized hAQP10 in the adipose tissue through selective immunolabeling (Fig. 1d). Thus, our data suggest that adipocyte glycerol flux augmented at lower pH associated with e.g. lipolysis is mediated by hAQP10.

**Architecture of hAQP10 and the glycerol-specific gate.** To resolve how hAQP10 is pH-gated, structural studies were initiated using polyhistidine (His)-tagged protein. However, full-length hAQP10 yielded no crystals and we continued with a variant (hAQP10$_{cryst}$; Fig. 2a) truncated in the termini (Δ1–10, Δ278–301) that crystallized at pH 6.0 (Supplementary Fig. 4). The structure was determined at 2.3 Å resolution (Table 1), and reveals a tetramer fold highly reminiscent of other AQPs (Fig. 2b), with each monomer formed by six transmembrane helices (TM1–TM6) establishing a conducting channel (Fig. 2c). Strikingly, the region typically linked to selectivity in AQPs, the aromatic and arginine (ar/R) selectivity filter at the non-cytosolic end of the pore[21], is significantly wider (2.6 Å) than in previously structurally characterized AQPs (Fig. 2c, d), as also found by HOLE analyses[22]. Furthermore, no glycerol molecule was identified at the ar/R filter, in contrast to the only available structure of a eukaryotic aquaglyceroporin, PfAQP (Fig. 2c, d)[17]. These observations indicate that the functional role of the ar/R region may not be maintained in hAQP10 (the ar/R filter of orthodox aquaporins is typically very tight, excluding glycerol passage, whereas in aquaglyceroporins it also permits flow of certain small solutes, but prevents most other compounds (Supplementary Fig. 1)[23]. A single glycerol molecule is instead located adjacent to the AQP archetypical, central NPA-motif (N82–A84) in hAQP10 (Fig. 2c, e, Supplementary Fig. 4b). This area and the presence of NPA glycerol molecule are highly conserved elements among structurally determined aquaglyceroporins (including GlpF and AqpM; Supplementary Figs. 1 and 5)[15,16]. However, toward the cytoplasm the glycerol molecule is rather positioned close to the unique F85 of loop B in hAQP10 (invariant as a valine/isoleucine in other AQPs). F85 has a side-chain configuration unfavorable for glycerol passage, forming a unique mechanistic feature. Moreover, the entire cytoplasmic pore region has adapted a tight arrangement not previously observed, achieved by the first part of loop B (G73-H80; loop layout likely allowed by the hAQP10-specific G73G74-motif; Fig. 2e and Supplementary Fig. 1) with V76-S77 capping the cytoplasmic opening, F85, and R94 (of TM3), which seemingly stabilizes loop B in the closed configuration. Notably, HOLE analysis suggests that this narrowing (0.9 Å) permits water (0.8 Å at the ar/R filter in hAQP2)[22], but not glycerol (1.3 Å in AqpM)[16] flux, in agreement with the proteopolymersome data at relatively high pH. The most likely pH-sensor candidate in the region is H80, which lines the pore and structurally links loop B, F85 and R94. pH-dependent gating mechanisms have been proposed for human AQP3, 4 and 5 refs[18,24,25], but the available structures are open (for hAQP4 the H80 equivalent was pinpointed as a gate, but loop B in hAQP4 has a significantly different configuration as compared to hAQP10)[26,27]. Thus, our findings pinpoint how human aquaglyceroporins may be gated at the intracellular interface[28], and how ligand-selective regulation may be achieved for a membrane protein.

**Functional characterization of hAQP10.** To assess the functional role and physiological importance of the cytoplasmic gate we investigated the functionality of hAQP10 and mutant forms using protepolymersome- (in vitro) and *S. cerevisiae*-based (in-vivo) assays (Supplementary Figs. 3b and c)[29]. In the reconstituted system, His-tag-fusions were assayed (Fig. 3a and

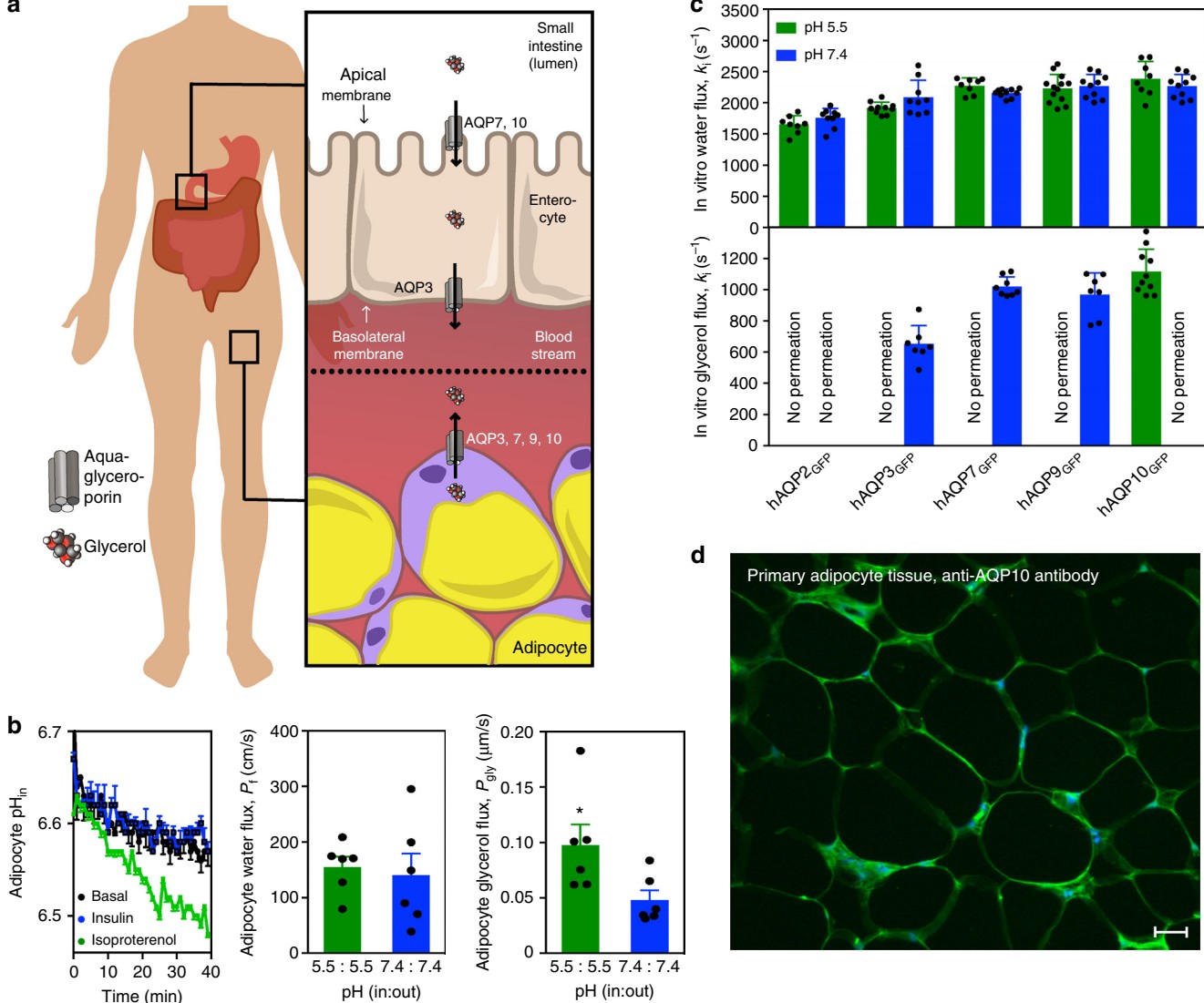

**Fig. 1** Low pH stimulates human adipocyte glycerol flux through aquaglyceroporin AQP10. **a** Simplified overview of aquaglyceroporin-mediated regulation of human body glycerol homeostasis. Glycerol absorption in the small intestine (enterocytes) occurs through AQP7 and 10, and via AQP3-mediated excretion into the bloodstream, whereas release into the circulation from fat tissue (adipocytes) involves AQP3, 7, 9 and 10. **b** Intracellular pH changes in human adipocytes under basal (control, black), lipogenic (insulin, blue) and lipolytic (isoproterenol, green) conditions. Results are given as mean ± SEM. $P$ < 0.001 isoproterenol $vs.$ control and insulin (ANOVA followed by Newman–Keuls's Q test; $N = 3$). Water and glycerol permeability of human adipocyte plasma membrane vesicles exposed to glycerol gradient. Flux was measured using identical pH inside and outside: pH 7.4 (blue) or 5.5 (green). Water ($P_f$) and glycerol ($P_{gly}$) permeability coefficients were calculated as described in Methods. Results are given as mean ± SEM. *$P = 0.037$ vs. 7.4:7.4 (Student's $t$-test; $N = 6$). **c** Water and glycerol permeability of GFP-fused human aquaporins reconstituted into polymersomes. $k_i$ rate constants ($s^{-1}$) were obtained at pH 7.4 (blue) and pH 5.5 (green). Each bar shows mean ± SD of $N = 7$–13 measurements performed for the same proteopolymersome sample. Data for shrinking proteopolymersomes indicate lack of glycerol flux and are not shown. See Supplementary Table 2 for a summary of the activity. **d** AQP10 is membrane-localized to subcutaneous human adipose tissue used for vesicle preparation. Representative immunofluorescence confocal microscopy images with anti-hAQP10 antibody (green) and DAPI staining for nuclei (blue). Scale bar: 200 μm

Supplementary Table 2), revealing an overall similar pH-dependency permeation profile for water and glycerol as for GFP-fused counterparts (the crystallized variant, hAQP10$_{cryst}$, mimics full-length protein, hAQP10). Furthermore, our experiments unambiguously pinpoint H80, F85 and R94 as critical for glycerol flux. Subsequently, we measured glycerol ($P_{gly}$) and water ($P_f$) flow rates in yeast cells challenged with osmotic gradients (Fig. 3b, c, Supplementary Fig. 6, Supplementary Tables 3 and 4). As in the proteopolymersome assay (Figs. 1c and 3a), opposing pH effects were found for hAQP3$_{GFP}$ and hAQP10$_{GFP}$, increasing glycerol permeation at low pH for hAQP10. Most importantly, the detrimental effect of H80A in vitro was reproduced,

suggesting a channel remaining closed independently of pH, in agreement with a pH-sensory role of the histidine; scouting prove that the H80A mutant is impermeable to glycerol over large internal pH and temperature spectra (Fig. 3b, Supplementary Fig. 6, Supplementary Tables 3 and 4). Impaired glycerol flux was also observed for mutations of H80-interaction network residues, S77 and R94 (both profoundly affected) and F85A (moderately influenced). Similarly, substitutions of G73, located distal to the channel, to valine and phenylalanine as in hAQP1 and 4, respectively, markedly reduced glycerol flux, supporting that the G73G74-motif is important to kink TM2, thereby allowing loop B to gate. In contrast, water-conducting flux remained unaffected

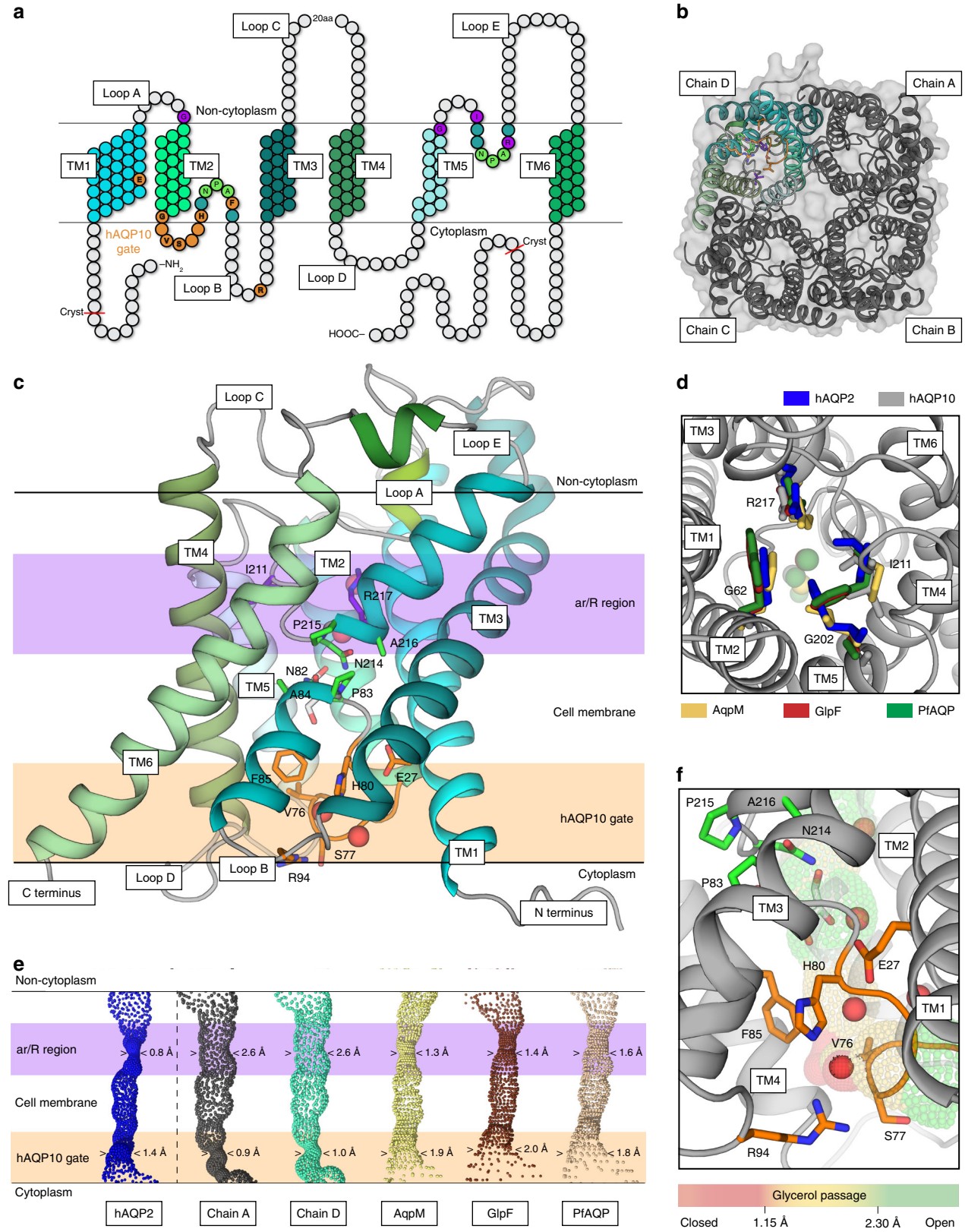

in vivo for almost all hAQP10 forms (Supplementary Fig. 6e). This is congruent with the proteopolymersome data (Figs 1c and 3a), substantiating that water diffusion through hAQP10 is pH-insensitive, maintained independently of the residues orchestrating glycerol flux. The effects of G73A and F85V are exceptional, displaying increasing flux of glycerol and water only at low pH, indicating that these subtle alterations (mimicking the case in hAQP3, 7 and 9, but likely with remaining differences in the microenvironment) maintain the pH-sensitive gating (without achieving complete closure for G73A).

**Fig. 2** Architecture of human AQP10 and the glycerol-specific gate **a** Topology of hAQP10 monomer with six transmembrane helices (TM1-6) and five connecting stretches (loops A–E). Residues at the NPA-motifs, the classical ar/R selectivity filter and the cytoplasmic gate are indicated in green, purple and orange (throughout); hAQP10-specific residues in bold. The length of the crystallized form is also highlighted. **b** The hAQP10 tetramer from the cytoplasmic side, with chains A–C shown in gray and chain D in cyan tones. **c** Side-view of the primed-to-open monomer (chain D). A single glycerol (sticks) and four water (red spheres) molecules were identified. **d** The unusually wide ar/R selectivity region of hAQP10 (chain A, gray) compared to those in hAQP2 (blue, pdb-id 4NEF)[22], AqpM (yellow, pdb-id 2F2B)[16], GlpF (brown, pdb-id 1FX8)[68] and PfAQP (wheat, pdb-id 3C02)[17]. View from the non-cytoplasmic side. Glycerol molecules in the structures are shown as spheres in equivalent colors. **e** The channel profiles of selected aquaporins calculated using the software HOLE. hAQP10 chains A (gray) and D (cyan) are compared with increasing minimal diameter from left to right. The cytoplasmic gate and ar/R regions are marked in light orange and purple, respectively. **f** Close-view of the cytoplasmic and glycerol-specific gate. H80 forms an interaction network work with E27, F85, R94, V76 and S77

## Table 1 Data collection and refinement statistics (molecular replacement)

|  | AQP10 |
|---|---|
| *Data collection* |  |
| Space group | $P2_12_12_1$ |
| Cell dimensions |  |
| *a, b, c* (Å) | 97.1, 116.8, 138.5 |
| α, β, γ (°) | 90.0, 90.0, 90.0 |
| Resolution (Å) | 50–2.30 (2.44–2.30)* |
| $R_{merge}$ | 11.9 (113.0) |
| $I / \sigma I$ | 12.5 (1.6) |
| Completeness (%) | 99.6 (97.6) |
| Redundancy | 7.84 (7.62) |
| *Refinement* |  |
| Resolution (Å) | 50–2.30 (2.34–2.30) |
| No. of reflections | 69920 |
| $R_{work} / R_{free}$ | 18.9/21.3 (30.3/34.8) |
| No. of atoms |  |
| Protein | 7493 |
| Ligand/ion | 372 |
| Water | 197 |
| B-factors (Å²) |  |
| Protein | 49.9 |
| Ligand/ion | 71.6 |
| Water | 56.0 |
| R.m.s. deviations |  |
| Bond lengths (Å) | 0.006 |
| Bond angles (°) | 0.891 |

*Values in parentheses are for highest-resolution shell

**hAQP10 gate opening mechanism**. How then is hAQP10 opened at low pH? Crystallization attempts at lower pH to structurally decipher the opening mechanism were fruitless. Nevertheless, as the obtained crystal form contains the entire hAQP10 tetramer in the asymmetric unit, intermonomeric differences were analyzed. While overall highly similar, three chains (Fig. 2b dark gray and Supplementary Fig. 7c) display an identical, closed, cytoplasmic arrangement. In contrast, a subtle shift of the pore-width (to 1.0 Å) is observed at H80 in chain D (Fig. 2b and d cyan). Hence, monomer D may represent a primed-to-open gate configuration (glycerol flux remains unanticipated at this pore-width). Equivalent examination reveals that the observed pore closure likely cannot be attributed to crystal packing or associated detergent molecules, as these interaction patterns differ between monomers (Supplementary Fig. 7). To unravel the molecular mechanism required for full opening we turned to molecular dynamics (MD) simulations of membrane-embedded hAQP10 tetramer in the presence of glycerol, assessing two different protonation states of H80 (mono(ε) and double), as a mimic of relatively high and low pH, respectively. We performed a cluster and principal component analysis on residues adjacent to H80, identifying four core groups describing

cytoplasmic pore configurations (Fig. 4a and Supplementary Fig. 8). The most frequent cluster (#1) resembled hAQP10 monomers (A–D) and was dominant for the monoprotonated simulation. This cluster was initially present for all monomers for mono and double protonation states, but the latter rapidly shifted to progressively more open pore arrangements (clusters #3–5, see also Supplementary Fig. 9). HOLE analysis of representative structures revealed a pore-width of 1.5 Å of cluster #4, sufficient to conduct glycerol (see also Supplementary Fig. 10). Notably, the simulations frequently displayed a non-single water file in the ar/R region and rather water coordination near the hAQP10 gate.

## Discussion

All-in-all, based on structural, functional and MD simulation analyses, we propose a pH-dependent gating mechanism of hAQP10 triggered by protonation of H80 (correlating with an increasing pKa value of this residue from closed to open, pKa from 3.6 to 7.1, respectively), which at low pH reorients (from chain A, D and cluster #1), stabilized by E27 (clusters #3–5) (Figs. 3c and 4a). With this structural shift, F85 adapts a more open side-chain orientation, and the loop (including V76–S77) rearranges in conjunction with R94 to allow glycerol permeation.

The present findings shed light on a key component of fat metabolism—how glycerol levels in the body are expected to be maintained through hAQP10-mediated influx (small intestine) and efflux (adipocyte tissue) (Figs. 1a and 4b). Glycerol flux across plasma membranes of adipocytes (and likely duodenal enterocytes, where reported pH was shown to be acidic) is demonstrated to be stimulated by low pH and unarguably linked to hAQP10, a protein previously shown to be highly physiologically relevant for glycerol flow in these cell types[10,30,31]. The determined hAQP10 structure represents a paradigm shift for future studies of aquaglyceroporins. Our combined analyses reveal that pH regulation is achieved by a cytoplasmic, glycerol-specific gate and, likely, a widened ar/R filter, both unique to hAQP10, correlating with intracellular acidification of adipocytes observed during lipolysis[5]. Thus, hAQP10 has potential for therapeutic intervention of obesity and metabolic diseases, as targeting the pH gate to allow constitutively high efflux of glycerol may prevent accumulation of TAGs inside adipocytes.

## Methods

**Plasmids, site-directed mutagenesis and yeast strains**. Codon-optimized human aquaporin cDNAs were purchased from GenScript, USA. Yeast-enhanced GFP was PCR amplified using AccuPol DNA polymerase (Amplicon, Denmark) and yeast codon-optimized version as template[32]. Supplementary Table 1 summarizes the plasmids and yeast strains used in this study. Briefly, for proteopolymersome reconstitution, each aquaporin was C-terminally fused to either a Tobacco etch virus (TEV) protease cleavage site attached to GFP and deca-histidine ($His_{10}$) tag (yielding e.g., $hAQP10_{GFP}$), or to an octa-histidine ($His_8$) stretch only (hAQP10, $hAQP10_{cryst}$ and hAQP10-derived mutant forms). Site-directed mutagenesis was performed by PCR[33] and the final DNA constructs were assembled in yeast by homologous recombination (see below). For crystallization studies, the $hAQP10_{cryst}$ variant was derived from hAQP10 by removal of the first 10 (N-terminal) and last 24 (C-terminal) amino acids, respectively. All expression

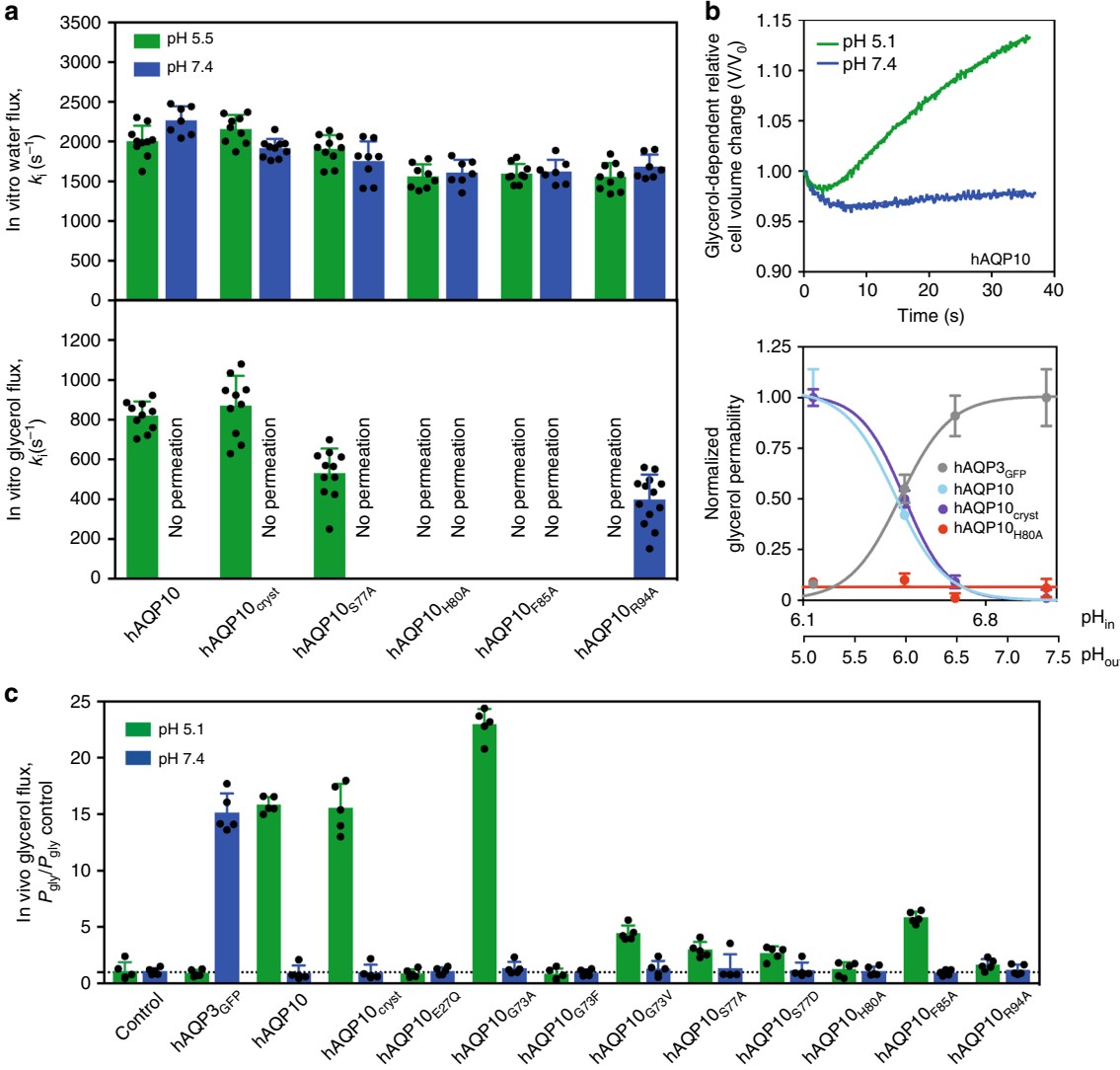

**Fig. 3** Functional characterization of human AQP10. **a** Water and glycerol permeability of hAQP10 forms reconstituted into polymersomes. $k_i$ rate constants ($s^{-1}$) were obtained at pH 7.4 (blue) and pH 5.5 (green). Each bar shows mean ± SD of $N = 7$–13 measurements performed for the same proteopolymersome sample. Data for shrinking proteopolymersomes indicate lack of glycerol flux and are not shown. See Supplementary Table 2 for a summary of the activity. **b** Upper plot: Representative time course of the relative cell volume ($V/V_0$) changes after glycerol osmotic shock at pH 5.1 (green) and 7.4 (blue) in hAQP10 expressing yeast cells. Lower plot: pH-dependence of glycerol permeability ($P_{gly}$) measured in cells expressing hAQP3$_{GFP}$ (gray) or different hAQP10 forms (cyan, blue and red). $P_{gly}$ is normalized for each dataset (($P_{gly}-P_{gly\ control}$)/$P_{gly\ max}$) and fitted with a Hill equation. Corresponding internal pH (pH$_{in}$) is also shown (lower axis). Results are given as mean ± SD of at least $N = 3$ independent experiments. **c** Glycerol permeability ($P_{gly}$) ratio of yeast cells expressing hAQP3$_{GFP}$ or hAQP10 forms measured at pH 5.1 (green) and pH 7.4 (blue). Results are normalized to $P_{gly}$ of the control strain at the respective pH. Data for hAQP10$_{F85V}$ are not shown due to the uncertainty of precise $P_{gly}$ estimation at pH 5.1 ($P_{gly}/P_{gly\ control} > 30$). See Supplementary Fig. 6e for the equivalent water flux data. Results are given as mean ± SD of $N = 5$ independent experiments

plasmids were assembled directly in the *S. cerevisiae* production strain PAP1500 (originating from Pedersen laboratory) by homologous recombination of HindIII-, SalI- and BamHI-digested pPAP2259[34] and aquaporin PCR products in presence or absence of a GFP PCR product[35]. Functional characterization in intact yeast cells was performed with wild-type aquaporins (tag-free) expressed from the methionine repressible promoter in pUG35[36]. The plasmids were generated by homologous recombination directly in the *S. cerevisiae* assay strain YSH1770, silenced for endogenous aquaporins AQY1 and AQY2 (10560-6B MATa leu2::hisG trp1::hisG his3::hisG ura35-2 aqy1D::KanMX aqy2D::KanMX)[18,25]. YSH1770 strain was produced in Soveral laboratory from the parental *S. cerevisiae* 10560-6B strain (provided by Patrick Van Dijck, Katholieke Universiteit Leuven and Flanders Interuniversity Institute for Biotechnology VIB, Belgium). Briefly, PCR amplified aquaporin cDNA fragments were co-transformed into YSH1770 strain with BamHI-, HindIII- and SalI-digested pUG35 for synthetic cDNA-derived hAQP10 and its variants, or SpeI- and ClaI-digested pUG35 for genomic cDNA-derived hAQP3 and GFP PCR products yielding hAQP3$_{GFP}$ construct. The nucleotide sequence of all used constructs was verified by DNA sequencing.

**Measurements of pH and glycerol release in human adipocytes.** Subcutaneous adipose tissue was obtained from healthy donors during hip replacement surgery (3 females and 8 males, age 53–70 years) following overnight fasting. The body mass index of the donors ranged from 24.4 and 37.5 kg m$^{-2}$ (27.72 ± 3.45; mean ± SD, $N$ = 11). None of the subjects suffered from known metabolic or malignant diseases or were taking medications known to alter the adipose tissue metabolism. The conducted procedures were approved by the Institutional Review Board at "IRCCS Policlinico San Matteo Foundation" in Pavia, Italy, and in accordance with the Helsinki Declaration of 1975 as revised in 2008. Each patient gave written consent for participating in the study.

Intracellular pH changes were monitored using 2′,7′ - bis - (2- carboxyethyl) - 5- (and - 6) - carboxyfluorescein acetoxymethyl ester (BCECF-AM, Sigma, USA) as fluorescent indicator. Briefly, freshly isolated human adipocytes were loaded with 5 μM BCECF-AM in PBS at 37 °C for 40 min. Cells were washed twice and resuspended with non-buffered isotonic mannitol. Time course of pH changes upon hormonal stimulation of BCECF-AM-loaded adipocytes was measured using CLARIOstar microplate reader (BMG LABTECH, Germany). Two different conditions to mimic lipogenesis or lipolysis (vs. basal) were applied, respectively:

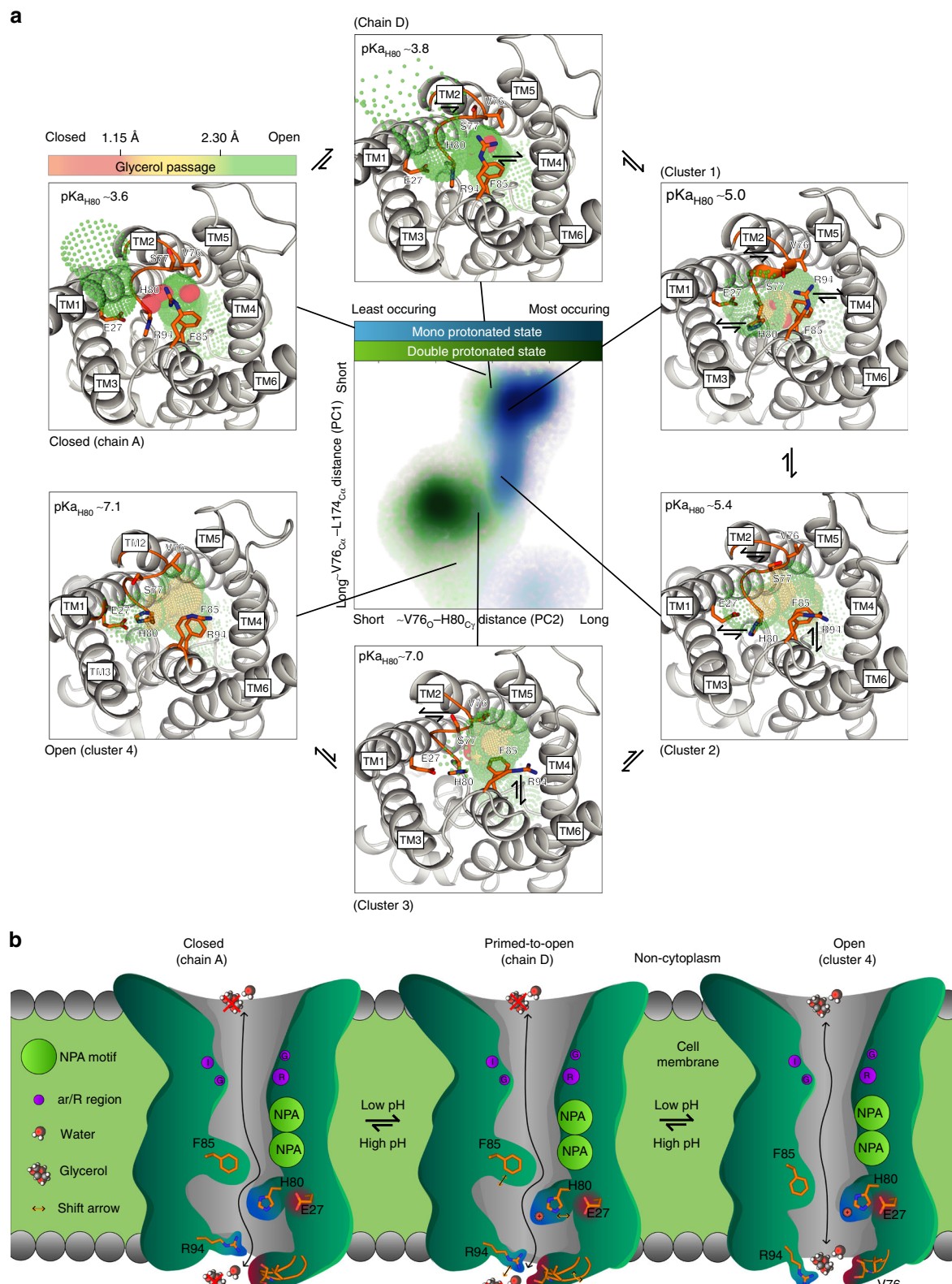

insulin (1 μM) or isoproterenol (50 μM) vs. control (untreated adipocytes). Each reaction started with the respective hormonal addition followed by the time course monitoring of pH changes by determining the ratio of fluorescence signals emitted at 530 nm when exciting at 490 and 440 nm (the isosbestic point of BCECF-AM), respectively. A calibration curve was determined to convert fluorescence measurements to the pH values. Briefly, BCECF-AM-loaded adipocytes were suspended in PBS at different pH (4.0, 6.0, 7.4 and 9.0) containing protonophore

carbonyl cyanide-4-(trifluoromethoxy) phenylhydrazone (FCCP, Sigma, USA) to equilibrate the intracellular pH with the extracellular medium. After 5-min incubation at 20 °C, the end-point fluorescence was measured as indicated above.

Measurements of glycerol release into the media were determined for the corresponding hormonal treatments described above. Briefly, freshly isolated human adipocytes were washed twice and resuspended with non-buffered isotonic mannitol followed by 20-min incubation at 20 °C. Subsequently, cells were exposed

**Fig. 4** Human AQP10 gate opening mechanism. **a** Cluster and principal component (PC) analysis of membrane-embedded hAQP10 molecular dynamics (MD) simulations with mono (mimicking a relatively high pH) and double (relatively low pH) protonation states of H80 yielding four main clusters (#1–4) of arrangements of the cytoplasmic gate region. A population distribution heat map of the principal components is shown in the central panel, with mono and double protonated frames shown in light blue-to-light blue and green-to-light green gradients, respectively. Simplified, PC1 and PC2 represent the pore-width distances between V76 and L174, and between V76 and H80 (see also Supplementary Figs. 8 and 9). Surroundings panels display crystal structure chains A and D, and representative structures of the MD-predicted clusters (#1–4), from closed to fully open with calculated HOLE profiles shown in traffic-light colors (red-to-green gradient). Indicative pKa$_{H80}$ values were calculated using PropKa. **b** Proposed hAQP10 pH-gated glycerol flux mechanism in adipocytes and likely other cell types. Glycerol, but not water, permeation is decreased at pH 7.4. AQP10 glycerol-specific opening is stimulated by pH reduction, triggering H80 protonation that renders the residue to interact with E27. Concerted structural changes of the nearby F85 and the cytoplasmic V76–S77 loop thereby allow glycerol passage

to the respective hormones and, at desired timepoints (ranging from 5 to 40 min), aliquots of suspension were taken. Hormonal treatments were terminated by heat shock (5 min, 100 °C). Amount of released glycerol was fluorometrically estimated ($\lambda_{ex} = 535/\lambda_{em} = 535$ nm) using glycerol assay kit (Sigma, USA) and employing CLARIOstar instrument. The amounts were normalized to the protein content of adipocytes suspension. Calibration curve using a glycerol standard solution was determined for each experiment.

**Transport in human adipocyte plasma membrane vesicles.** Adipocyte plasma membrane vesicles were prepared as previously described[37]. Briefly, 3–8 g of freshly excised adipose tissue was homogenized in an ice-cold buffer containing 10 mM Tris-HCl pH 7.4, 250 mM sucrose, 1 mM EDTA. The homogenate was then centrifuged (3000 × g, 15 min, 4 °C), the superficial solidified fat and pellet eliminated, and the infranatant centrifuged again (12,000 × g, 15 min, 4 °C). The resulting pellet consists of adipocyte plasma membrane vesicles, as assessed morphologically previously[10]. Water and glycerol permeabilities of isolated adipocyte plasma membranes were essentially measured exploiting stopped-flow light scattering as previously described[10]. Briefly, vesicles were suspended in solutions at pH 7.4 or 5.5 (10 mM KH$_2$PO$_4$/K$_2$HPO$_4$, 136 mM NaCl, 2.6 mM KCl) and incubated at RT for 30 min. Subsequently, vesicles were subjected to a 145 mM inwardly directed glycerol gradient (the solutions contained 10 mM KH$_2$PO$_4$/K$_2$HPO$_4$ buffer at pH 7.4 or 5.5). Initially there is an increase in light scattering resulting from vesicle shrinkage caused by osmotic water efflux (water flux), followed by slower decrease resulting from vesicle swelling caused by glycerol entry triggering influx of water (glycerol flux). The water permeability coefficient (P$_f$) was calculated from the following equation as previously described[38]: P$_f$ = k·V$_0$/ΔC·V$_w$·A, where ΔC is the osmotic gradient, V$_w$ the molar water volume, V$_0$ the cell volume and A the vesicle surface area. The glycerol permeability coefficient (P$_{gly}$) was calculated using the following equation: P$_{gly}$ = 1/[(S/V)τ], where S is the vesicle surface area, V the cell volume, and τ (K$^{-1}$) is the exponential time constant fitted to the vesicle swelling phase of the light scattering time course corresponding to glycerol entry[39]. Immunolocalization of hAQP10 in human adipose tissue was performed using anti-hAQP10 rabbit polyclonal affinity isolated antibody (1:300 dilution; SAB2103514, Sigma, USA) followed by incubation with AlexaFluor 488-conjugated goat anti-rabbit antibody (1:500 dilution; 111-546-046, Jackson ImmunoResearch Europe Ltd, UK) as previously described[10]. The fluorescent dye diaminophenyl-indole (*DAPI*; Molecular Probes, USA) was used to visualize nuclei. Slides were examined with a TCS SP5 II LEICA confocal microscopy system (Leica Microsystems, Italy) equipped with a LEICA DM IRBE inverted microscope. Negative controls (not shown) were performed by incubating slices with the non-immune serum.

**Protein production for polymersome assay and crystallization.** TEV-GFP-His$_{10}$- and His$_8$-fusions were produced essentially as previously described[34,35]. Briefly, a single colony of transformed PAP1500 cells was grown until stationary phase in 5 mL of glucose minimal medium supplemented with leucine and lysine. Subsequently, 200 µL of the culture was propagated in 5 mL glucose minimal medium supplemented with lysine. Next day, 1 mL of this culture was used to inoculate 50 mL of the same medium. The following day this pre-culture was used to inoculate 1 L of glucose minimal medium supplemented with lysine. The overnight culture was subsequently transferred to 10 L of amino acid-supplemented minimal medium containing 3 % glucose and 3 % glycerol as carbon source, and propagated in an Applikon bioreactor equipped with an ADI 1030 Bio Controller connected to a PC running the BioExpert software (all from Applikon, Holland) as described previously[34]. The initial part of the fermentation was performed at 20 °C. The bioreactor was fed with glucose to a final concentration of 2 % when the initial glucose had been metabolized. The pH of the growth medium was maintained at 6.0 by computer-controlled addition of 1 M NH$_4$OH. The shift from growth on glucose to glycerol was monitored as a decrease in the rate of NH$_4$OH consumption. At this point, the bioreactor was cooled to 15 °C and protein expression was induced by addition of galactose to a final concentration of 2 %. Cells were harvested 72 h post induction.

Protein was purified essentially as described previously[35]. Yeast cells were disrupted by glass bead homogenization (BioSpec, USA). Briefly, yeast cells were

resuspended in ice-cold lysis buffer (25 mM Tris-HCl pH 7.5, 500 mM NaCl, 20 % glycerol, 5 mM BME, 1 mM PMSF) supplemented with SIGMAFAST protease inhibitor cocktail (Sigma, USA). After mechanical disruption, cell debris was pelleted by centrifugation (3000 rpm, 20 min, 4 °C) and the membranes were isolated from the supernatant by ultra-centrifugation (205,000 × g, 3 h, 4 °C). Crude membranes were resuspended in solubilization buffer (20 mM Tris-HCl pH 7.5, 200 mM NaCl, 20 % glycerol, 5 mM BME, 1 mM PMSF) supplemented with SIGMAFAST protease inhibitor cocktail, homogenized in a Potter-Elvehjem homogenizer and stored at −80 °C until further use. Isolated membranes were solubilized in 2 % n-decyl-β-D-maltopyranoside (DM; Anatrace, USA) and each aquaporin was purified using immobilized metal affinity chromatography (IMAC). Briefly, detergent-solubilized material was clarified by ultracentrifugation (120,000 × g, 1 h, 4 °C), diluted 2 × in IMAC buffer (20 mM Tris-HCl pH 7.5, 200 mM NaCl, 20 % glycerol, 5 mM BME, 0.2 % DM) and filtered using a 0.45 μm filter. Each sample was then bound to a nickel-charged affinity HisTrap HP column (GE Healthcare, Denmark), and bound protein was eluted in IMAC buffer using an imidazole gradient. GFP-TEV-His$_{10}$- and His$_8$-tagged variants used for proteopolymersome reconstitution were produced from membranes solubilized in 0.5 % n-hexadecyl-phosphocholine (FC-16; Glycon Biochemicals, Germany) and following the binding eluted in IMAC buffer containing 3 % lauryldimethylamine-N-oxide (LDAO; Anatrace, USA) and not subjected to size exclusion chromatography (SEC). Top IMAC fractions of the crystallization variant (hAQP10$_{cryst}$) were pooled, concentrated using Vivaspin 20 concentrators (MWCO 100 kDa; Sartorius, Germany), and subjected to SEC using a Superdex increase 200 10/300 GL column (GE Healthcare) equilibrated in SEC buffer (20 mM Tris-HCl pH 8, 100 mM NaCl, 10 % glycerol, 2 mM BME, 0.4 % n-nonyl-β-D-glucopyranoside (NG; Anatrace, USA)).

**Functional characterization in proteopolymersomes.** Poly (2-methyloxazoline)-block-poly (dimethyl siloxane) di-block copolymer PDMS$_{34}$PMOXA$_{11}$ (PDMS-PMOXA; DSM, Denmark) polymersomes were prepared by the co-solvent method as previously described[40–44]. Briefly, 15 mg of PDMS-PMOXA copolymer was dissolved in 50 µL ethanol and added dropwise to 4450 µL of 10 mM PBS pH 7.2, 136 mM NaCl, 2.6 mM KCl, followed by 24-h dialysis against PBS with 3 exchanges of the buffer. Proteopolymersomes were prepared in a similar manner where 15 mg of dissolved PDMS-PMOXA copolymer was mixed with PBS containing 25 µg of the respective purified aquaporin sample. After dialysis, all samples were extruded 15 times through a 200 nm polycarbonate filter (Whatman, USA). The dimensions of the extruded vesicles (hydrodynamic diameter) were determined at 20 °C by dynamic light scattering (DLS) using ZetaSizer NanoZs instrument (Malvern, UK). The water flux was measured employing a Bio-Logic SFM 300 stopped-flow device (Bio-Logic, France), with a monochromator at 517 nm and a cutoff filter at 530 nm, respectively. For each individual stopped-flow test, 0.13 mL of extruded polymersomes or proteopolymersomes was quickly mixed with 0.13 mL of 0.5 M NaCl, which caused the vesicles to shrink due to osmotically driven water efflux. At least 7 tests were performed for each sample; the dead time for the mixing of stopped-flow injection was 5 ms. Vesicle size changes were monitored and recorded in the form of an increasing signal in the DLS analysis. Obtained kinetic data were fitted with a single exponential equation, and the rate constant (s$^{-1}$) that is directly proportional to the water flux through the polymeric membrane was determined using Origin software (OriginLab Corporation, USA). In the glycerol transport assay 3 mL of extruded polymersomes was incubated with 3 mL of 2 M glycerol overnight at 4 °C to mediate glycerol transport into the polymeric vesicles. After incubation, the dimensions of the polymeric vesicles were determined by DLS. Glycerol flux in proteopolymersomes was assessed using stopped-flow after mixing the samples with 0.5 M of NaCl (exhibiting the same osmotic pressure as 1 M glycerol). The summary of the obtained activity data is listed in Supplementary Table 2.

**Crystallization and structure determination.** hAQP10$_{cryst}$ crystals were grown by hanging-drop vapor diffusion at 18 °C by mixing protein solution (~ 4 mg mL$^{-1}$) supplemented with 0.3 mM n-nonyl-β-D-thioglucoside (Hampton Research, USA) with a reservoir solution composed of 100 mM MES-monohydrate-NaOH pH 6.0, 19 % PEG 2k MME, 5 % glycerol and flash frozen in liquid nitrogen. X-ray

diffraction data were collected using an EIGER detector at the Paul Scherrer Institut, Villigen, Switzerland, beam line X06SA. Data processing was done using the software XDS[45]. Crystals belonged to space group P2₁2₁2₁ with cell dimensions a = 97.1 Å, b = 116.8 Å, c = 138.5 Å. The initial phases were determined by molecular replacement with software PHASER using *E. coli* glycerol facilitator (GlpF) structure (pdb-id 1LDF[46]) yielding an entire tetramer in the asymmetric unit. Model building and refinement were done using COOT[47] and phenix.refine[48] iteratively. TLS refinement was introduced in the final refinement rounds[49]. The final refinement statistics are listed in Table 1. All structure figures were generated using Pymol.

**HOLE analysis of the pore dimensions**. The software HOLE (version v2.2.005) was obtained from www.holeprogram.org[50]. Pore profiles were analyzed until the radius reached 5 Å and van der Waals radii were subsequently determined. The pore profiles from MD simulations were analyzed without the placement of hydrogen atoms to make the investigation comparable with the pore profiles in the crystal structure. Analysis was performed after removal of water molecules and hetatoms with passage through S77, H80, and R94, and the pores colored according to the water permeability.

**Functional characterization in yeast cells**. YSH1770 strain (described in Plasmids, site-directed mutagenesis and yeast strains section) was grown at 28 °C with orbital shaking in YNB (yeast nitrogen base) without amino acids (DIFCO), with 2 % (w/v) glucose and supplemented with the adequate requirements for prototrophic growth. Transformants were grown to OD₆₄₀nm ≈ 1 (corresponding to 1 × 10⁷ cells mL⁻¹), harvested by centrifugation (5000 × g, 10 min, 4 °C), washed three times and resuspended in ice-cold sorbitol (1.4 M) K-citrate buffer (50 mM pH 5.1 or pH 7.4) up to a concentration of 0.33 g (wet weight) mL⁻¹, and kept on ice for at least 90 min. Prior to the osmotic challenges, the cell suspension was preloaded with the nonfluorescent precursor 5-and-6-carboxyfluorescein diacetate (CFDA, Sigma, USA; 1 mM for 20 min at 30 °C) that is cleaved intracellularly by non-specific esterases, and generates the impermeable fluorescent form (CF) known to remain in the cytoplasm[51]. Cells were then diluted 1:10 in 1.4 M sorbitol buffer and immediately used for stopped-flow experiments.

Equilibrium cell volumes (V₀) were obtained by loading cells with CFDA under an epifluorescence microscope (Zeiss Axiovert, Zeiss, Jena, Germany) equipped with a digital camera as previously described[51]. Cells were assumed to have a spherical shape with a diameter calculated as the average of the maximum and minimum dimensions of each cell. Stopped-flow experiments were performed on a *Hi-Tech* Scientific PQ/SF-53 apparatus (*Hi-Tech* Scientific, UK) with 2 ms dead time, temperature-controlled, interfaced with a microcomputer. Permeability assays were performed at 23 °C, except for activation energy (Ea) assays where temperature ranged from 10 to 34 °C. Five runs were usually analyzed in each experimental condition. In each run 0.1 mL of cell suspension (1:10 dilution in the resuspension buffer) was mixed with an equal amount of iso (baseline) or hyperosmotic solution (sorbitol or glycerol 2.1 M, 50 mM K-citrate buffer pH 5.1 or pH 7.4) of 1.25 tonicity ((Λ = (osm_out)∞/(osm_out)₀)). Fluorescence was excited using a 470 nm interference filter and detected using a 530 nm cutoff filter. The time course of cell volume change was followed by fluorescence quenching of the entrapped fluorophore (CF). The fluorescence traces obtained were corrected by subtracting baseline (reflecting the bleaching of the fluorophore). The calibration of the resulting traces was performed followed our previous strategy[52], where a linear relationship between relative volume and F was obtained (v_rel = a F/F0 + b), and the values of a and b were estimated individually for each osmotic shock. The permeability coefficients for water (P_f) and glycerol (P_gly) transport were evaluated using the analysis described previously[53]. The calibrated experimental data were fitted to theoretical curves, considering the water and glycerol fluxes and the resulting changes in cell volume and intracellular concentrations of solutes. Optimization of permeability values was accomplished by numerical integrations using the mathematical model implemented in the Berkeley Madonna software (http://www.berkeleymadonna.com/). Estimations of the internal pH (pH_in) were performed as previously described[54]. The activation energy (Ea) of glycerol permeation was evaluated from the slope of the Arrhenius plot (ln P_gly as a function of T⁻¹) multiplied by the gas constant R.

**MD simulations**. MD simulations were performed on the tetrameric hAQP10_cryst structure embedded in a palmitoyloleoyl-phosphatidylethanolamine (POPE) bilayer. Two systems were built with the imidazole ring of H80 monoprotonated at ε-nitrogen or double protonated, to mimic relatively high and low pH, respectively. The detergent molecules, the three intermediate glycerol molecules located between the monomers and water molecules on the hydrophobic exterior side of the protein were removed from the system, while water molecules present inside the protein were kept. Missing atoms were added using the software VMD[55] with the PSFGEN plugin. Side chains were kept at their default protonation state including remaining H residues (protonated at ε-nitrogen). Additional water molecules were placed using the software DOWSER[56], according to an energy threshold of −12 kcal mol⁻¹. The protein was aligned in the XY plane using the VMD plugin ORIENT, and was subsequently solvated using the program SOLVATE[57]. A partially hydrated POPE membrane of 127 Å × 127 Å bilayer patch was built using the VMD plugin

MEMBRANE and aligned to the hydrophobic part of the protein. Lipids overlapping with the protein were removed to avoid steric clashes. The final solvation of the system was done by adding two 15 Å layers of water while water molecules in the hydrophobic part of the membrane were removed. To model the protein in natural ionic concentrations, the system was electroneutralized at 150 mM of Na⁺ ions using the VMD plugin AUTOIONIZE. The H80 ε-nitrogen-protonated system consisted of 15177 protein atoms, 330 POPE lipid molecules, 18549 water molecules, and 52 Na⁺, 59 Cl⁻ ions for a total of 112241 atoms. The double protonated system consisted of 15,181 protein atoms, 326 POPE lipid molecules, 18490 water molecules, and 52 Na⁺, 63 Cl⁻ ions for a total of 111,572 atoms.

Four cases for each protonation state of H80 were simulated with ACEMD[58] with the CHARMM27 parameter set[59,60] using TIP3P for water molecules[61]. Glycerol was modeled with the CHARMM36 parameter set[62]. Each simulation was initially minimized for 5000 steps and then simulated for 50 ps in the NVT ensemble followed by 500 ps in the NPT ensemble with all protein, lipid and glycerol molecules restrained with a spring constant of 1 kcal mol⁻¹ Å⁻². Next, all restraints on the protein and lipids were removed, while each glycerol molecule was constrained below the respective NPA-motif (residues 82–84). The constraint was handled by the PLUMED plugin[63] and calculated by the distance between the center of mass of glycerol and the backbone atoms of residues 82–84 using a harmonic force constant of 1 kcal mol⁻¹ Å⁻². The simulations were then run for 120 ns in the NPT ensemble.

The simulations were carried out at constant temperature (310 K) and pressure (1 atm) using Langevin dynamics with a damping coefficient of 1 ps⁻¹. The Particle Mesh Ewald method[64] was used for evaluating electrostatic forces with a resolution of at least 1 Å. A cutoff of 10 Å using a switching function beginning from 8.5 Å was employed. The integration time step was 2 fs and short- and long-range electrostatic interactions evaluated every 2 fs and 4 fs, respectively. Analysis of the trajectories was performed using the software VMD[55]. During the simulations, the root mean square displacement (RMSD) on the protein backbone atoms and the area per lipid were monitored to ensure stable physical behavior. The total RMSD remained below 2.0 Å² while the area per lipid equilibrated toward 55 Å² (see Supplementary Figs. 4a and b for two representative simulations).

Clustering and principal component analysis (PCA) was performed using CPPTRAJ[65] on the combined trajectory of the four monomers for all simulation cases. As all monomers were put in sequence, the analysis is based on 3.840 μs cumulative simulation time. Both analyses were performed on CA atoms of 28 selected residues near the cytoplasmic region of the loop. The PCA was carried out on 192,036 frames (6001 structures per monomer and four crystal structures), corresponding to every 10,000 simulation step. The clustering was then performed on the same dataset using the average linkage K-MEANS algorithm[66] and sieving every fifth frame. Ten clusters were identified with the criteria that the cluster centroid should be located on the most populated locations on the PCA heat map (Fig. 4a).

PropKa3.1 was used to determine the pKa values of H80 in monomers of the crystal structure and the different clusters generated in MD simulations[67]. The input pdb files were prepared by removing hetatoms and other chains prior to pKa analysis. Structures obtained from MD simulations were devoid of hydrogen atoms to resemble the crystal structure. Glycerol molecules were also omitted in the crystal structure to ensure that the calculations were comparable with the MD simulations.

**Data availability**
Data supporting the findings of this manuscript are available from the corresponding authors upon reasonable request. Atomic coordinates and structure factors for the human AQP10 crystal structure have been deposited at the Protein Data Bank (PDB) under accession code 6F7H which will be available upon publication. Readers are welcome to comment on the online version of the paper.

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

## Acknowledgements

K.G. is supported by The Independent Research Fund Denmark and The Lundbeck Foundation. P.G. is supported by the following Foundations: Lundbeck, Knut and Alice Wallenberg, Carlsberg, Novo-Nordisk, Brødrene Hartmann, Agnes og Poul Friis, Augustinus, Crafoord as well as The Per-Eric and Ulla Schyberg. Funding is also obtained from The Independent Research Fund Denmark, the Swedish Research Council and through a Michaelsen scholarship. S.F.T. and C.H.N. are supported by the IBISS and MEMENTO grants from Innovation Fund Denmark. P.A.P is supported by the following foundations: The National Danish Advanced Technology Foundation, The Strategic Research Council, The Independent Research Fund Denmark, and Novo-Nordisk. A.F.M. and G.S. are supported by Fundação para a Ciência e Tecnologia, Portugal (SFRH/BD/52384/2013 and UID/DTP/04138/2013). We are thankful to David Sørensen for technical assistance, Nanna Louise Sørensen for help with figures and Kresten Lindorff-Larsen for support with the MD simulations. We are grateful for assistance with crystal screening and data collection at the Swiss Light Source, the Paul Scherrer Institute, Villigen, beam line X06SA. Access to synchrotron sources was supported by The Danscatt program of The Independent Research Fund Denmark.

## Author contributions

K.G. performed protein purification and crystallization from cell material provided by S. K. and P.A.P. K.G. and P.G. collected X-ray diffraction data and solved the structure. J.W.M. prepared His-tagged protein samples for polymersome reconstitution guided by P.A.P. P.A.P supplied purified GFP-tagged AQPs for functional characterization in yeast. P.A.P. supplied all mutants. A.F.M. performed activity characterization in intact yeast cells steered by G.S. S.F.T. conducted MD simulations supervised by C.H.N. M.S. performed activity characterization in proteopolymersomes and P.A.P. analyzed all proteopolymersome data. K.W. refined the crystal structure. J.W.M. performed structural analysis. U.L. performed activity characterization in adipocyte plasma membrane vesicles and immunofluorescence confocal microscopy. K.G., P.A.P. and P.G. designed the project. K.G. and J.W.M. generated figures, K.G. and P.G. wrote the paper with contribution from all authors.

## Additional information

**Competing interests:** S.K., M.S., C.H.N., and P.A.P. have interests in Aquaporin A/S. The remaining authors declare no competing interests.

