## [Peer Review File · Nature Communications]

Reviewers' Comments:

Reviewer #1:

Remarks to the Author:

The authors describe a novel crystal structure of a human aquaglyceroporin, AQP10. It is the first structure of a mammalian aquaglyceroporin, with the structure as well as functional data suggesting an intriguing pH gating. The findings are interesting and relevant not only for structural biologists and biophysicists, but also from a clinical point of view: AQP10's role in glycerol metabolism makes it a potential target in obesity treatment. The paper is well written and easy to follow, and the findings are relevant. However, particularly the molecular dynamics simulation data are inconclusive and require extensive revision before I can recommend the paper for publication.

First, throughout the manuscript the pore radius is used as a proxy for permeation, with a radius of $> 0.8\text{\AA}$ presumed to be water permeable and a radius of $> \sim 1.2\text{\AA}$ presumed to be glycerol permeable. Why aren't permeabilities in MD used to validate this assumption? Is the crystallographic (closed) state indeed permeable to water, but not glycerol? Likewise, how is this different in the putatively open state? At least a glycerol permeation profile should be computed (and compared to e.g. GlpF) to be able to safely conclude that the putatively open state is indeed permeable to glycerol.

Second, the statistics of the observed opening remains unclear. Is the opening transition observed in all monomers, or only in some? Multiple transitions would be necessary for a statistically significant result.

Third, has the pKa of H80 been measured or estimated, to test the hypothesis of a doubly protonated H80? And how would the pKa of the other H80 shift if the first has been doubly protonated? Presumably in the low dielectric of the membrane an interaction between the positive charges on the doubly protonated His might occur. Therefore, how probable is the state with all four H80 doubly protonated?

Finally, how unique is the local environment around H80 for AQP10? Would it be feasible that other aquaporins or aquaglyceroporins share a similar mechanism?

Reviewer #2:

Remarks to the Author:

Gotfryd et al. present a crystal structure of the human aquaglyceroporin 10 at 2.3\AA resolution. Besides the prokaryotic GlpF from *Escherichia coli* and the eukaryotic PfAQP from *Plasmodium falciparum* this is the third structure of a water/glycerol channel. As a special feature of hAQP10, the authors point out that the size exclusion selectivity filter (aromatic/arginine region) is considerably wider than in the other aquaporins and does not hold a glycerol molecule in the crystal structure. Further, their structure appears in a closed configuration with regard to glycerol towards the cytoplasmic end of the channel, whereas the smaller water might still pass. Due to the presence of a histidine in this area, a pH gating mechanism is proposed that opens the channel at low pH. The authors further try to corroborate this interpretation by functional measurements of wildtype and mutant hAQP10 in cell- and vesicle-based assays at different pH conditions. There are, however, major inconsistencies here. The physiological context also does not fully fit into the scenario. Besides adipocytes where the pH may drop during lipolysis, AQP10 is prominently expressed in the small intestine and facilitates glycerol uptake at neutral pH. In adipocytes, why would pH-regulation of AQP10 be necessary (and kept during evolution) if a permanently open AQP10 would do the same trick and possibly even better (cattle obviously can dispense with

AQP10 completely because it is a pseudo gene; Tanaka et al. *Biochem Biophys Res Commun*. 2015 Mar 28; 1: 16-21).

Specific points:

Suggested pH regulation:

1. In a reconstituted polymersome system, the authors show water and glycerol permeability of the human aquaglyceroporins AQP3, AQP7, AQP9, and AQP10 at neutral pH (Fig. 1c). When shifting to pH 5.5, water permeability appears unaffected but glycerol permeability of AQP3, AQP7, and AQP9 ceases, while it increases dramatically (about one order of magnitude) with AQP10. The data do not match with earlier data. AQP3 (Zelenina et al. *J Biol Chem*. 2003 Aug 8; 278(32): 30037-43) have shown inhibition of AQP3 at low pH and identified involved residues at the extracellular face of the protein. Also recently AQP7 has been shown to be blocked by low pH from the extracellular side (Rothert et al. *J Biol Chem*. 2017 Jun 2; 292(22): 9358-9364). However, in the same study, glycerol permeability of AQP9 was insensitive to pH changes, i.e. AQP9 was functional down to pH 3. Therefore, AQP9 glycerol permeability should have been seen in the assay. Lack thereof renders the system and the data quite doubtful.

2. In the same figure (1b), vesicles prepared from adipocytes show only a mild, probably twofold increase in glycerol permeability at pH 5.5 with a weak statistic p-value of 0.037. Considering that AQP10 accounts for 50% of the glycerol flux at neutral pH (Laforenza et al. *PLoS One*. 2013; 8(1): e54474), a much stronger effect should have been seen to match the data from the polymersomes. Again, how does this fit?

3. pH-shifts in the yeast cell-based assays of AQP10 were done in the media, i.e. at the extracellular side, and not where the intracellular pH-gate is said to be located (or where the lipolysis-derived pH drop should occur in adipocytes). Still, a 4-6fold increase in glycerol permeability is reported by the authors (Suppl. Table 5) not fitting their proposed gating mechanism.

4. The AQP10 crystal structure was solved at pH 6.0, i.e. close to the pH of 5.5 at which the authors claim the channel to be open. However, they find a closed structure. Please explain.

5. Ishii et al. (*Cell Physiol Biochem*. 2011; 27(6): 749-56) even suggested a dual functional mechanism of AQP10 that requires trans-stimulation by smaller concentrations of glycerol at the lower end of the transmembrane gradient. This has not been considered in the assays or stimulations.

Wide ar/R region:

5. The feature of a particularly wide ar/R filter is not interpreted in a physiological context. Only a brief statement questioning "if the functional role of the ar/R region is maintained in hAQP10" is given.

Together, it is very much appreciated to have a new aquaglyceroporin structure. However, it remains unclear whether the structure provides new insight into aquaglyceroporin function because the concept of AQP10 pH-gating needs more consistency and substance.

Reviewer #3:

Remarks to the Author:

In this manuscript, Gotfryd et al. present the structural and functional studies on human

aquaglyceroporin AQP10 to provide the molecular basis of its pH-dependent regulation. The authors first show that glycerol flux into human adipocytes is pH-dependent, and pinpoint AQP10 as a responsible protein for such pH-dependent glycerol flux. The authors next describe the structure of human AQP10, determined at 2.3 Å resolution, which reveals the unique features of AQP10 as compared to other AQPs: 1) the wide ar/R selectivity filter, 2) the cytoplasmic restriction formed by F85, and 3) the loop B capping the cytoplasmic vestibule. Combined with structural, biochemical and computational analyses, the authors propose a pH-gated glycerol permeation mechanism of AQP10 where His80 and Phe85 play an important role.

The major achievement here is the structure determination of a human aquaglyceroporin; it is for the first time the structure of a mammalian aquaglyceroporin is presented. Although the structures of bacterial and parasite aquaglyceroporins have already been reported, the present structure will allow a more systematic comparison of aquaglyceroporins across different organisms. The structure also provides an unexpected finding; the loop B forms a constriction at the cytoplasmic vestibule, which is likely to be responsible for pH-gating. Although the data presented here are not sufficient to derive a strong conclusion on how pH-gating is achieved through this loop, the high-resolution structure nevertheless provides a basis for further functional and computational analyses on this aspect.

Overall, this is an important study that represents a significant advancement in understanding glycerol permeation in human cells through AQPs. I recommend this manuscript for publication in Nature Communications, pending some revisions as detailed below.

1) A major drawback of this manuscript is the lack of clear, logical explanation on how low pH, at which AQP10 opens, is related to fat metabolism in the physiological context. While the pH-gating mechanism of AQP10 itself is plausible, how such pH-dependent glycerol permeation impacts human fat diet is poorly discussed. Please provide more physiological insights from other literature, if any.

2) From the manuscript it is not clear whether the structure is of an open (glycerol-conductive) or a closed state. I suggest adding a more clear statement about the conductive state of the channel. The authors argue that the protonation of His80 is the central mechanism of the gating at cytoplasmic site, in the latter part of this paper. The reviewer thinks this is very interesting, and also thinks confirming the protonation state of the His80 in this crystal structure is important. The pKa of the histidine residue is 6.04, so in this crystallizing condition, it is possible that His80 is double protonated. In that case, the double protonated His80 will also stabilize the closed state. So the reviewer suggest that the authors should calculate the pKa of His80 and discuss about the protonation state of His80.

3) The omit electron density map in Supplementary Figure 3 shows that glycerol molecule is only partially visible. This raises question as to whether the observed density really belongs to a glycerol, or other molecules such as partially ordered water. Have the authors tried refinement by placing water molecules instead of a glycerol? Related to this, it is not clear whether this glycerol-binding site correspond to the previously observed glycerol-binding sites in other AQPs? I suggest adding a figure comparing the glycerol-binding sites of different AQPs.

4) Most structural description in the manuscript focus the residues on loop B, but they are poorly represented in the figures. I suggest including figures depicting the structural elements below.

- The position of the G73G74-motif.
- The distances between E27, H80, and R94 to show whether these residues are within the hydrogen-bonding distances. How F85 and R94 stabilize the loop B should be referred.
- The distance between F85 and glycerol.
- Other possible hydrogen-bonding or hydrophobic interactions in this region.
- Structural comparison of the loop B from AQP10 and different aquaporins.
- How the loop B interacts with the cytoplasmic gate is not shown in the Fig. 2-c. The close view of

the loop B will be needed to discuss the "tight arrangement".

5) The authors predict that the pH-sensitivity is missing in other human aquaglyceroporins because of the valine replacement of F85. The authors could test this possibility by measuring the pH-dependent glycerol flux of F85V mutant.

6) The authors argue that "All-in-all, based on structural, functional and MD simulation analyses, we propose a pH-dependent gating mechanism of hAQP10 triggered by protonation of H80, which at low pH reorients (from chain A, D and cluster #1), stabilized by E27 (clusters #2-4) (Fig. 4a)". In this argument, the electrostatic interaction between H80 and E27 caused by the protonation of H80. This is the major topic of discussion, so the substitution mutation analysis such as E27Q is essential.

7) The result and the discussion of the MD simulation seems to be consistent with those in the paper "ref 23 (S. Kaptan et al., H95 Is a pH-Dependent Gate in Aquaporin 4. Structure 23, 2309-2318(2015))". The reviewer thinks that the paper "ref 23" should also be discussed in the discussion section.

8) I would like to know how the proposed pH-gating mechanism of AQP10 compares to that proposed for other AQPs.

9) The reviewer thinks that the distribution of the distance between H80 and E27 in the MD simulation is very important, so these data (Supplementary Fig. 8a-b) should be shown in main Fig. 4.

Point-to-point response to the Reviewers' comments

We thank the editor for the positive evaluation of our paper, and the three reviewers for recognizing the merit of our work and for providing very valuable critique. As detailed below, we have addressed the Reviewers' comments fully and carefully, and revised the manuscript accordingly. To facilitate the reviewing process, we have copied the Reviewers' original comments, which are shown in **black**. Our responses are then shown in **blue**.

General comment:

Erroneously, in the list of References (main text of the manuscript), we previously listed an incorrect reference (under number 29). The correct one should be: H. Li *et al.*, Expression and localization of two isoforms of AQP10 in human small intestine. *Biol Cell* **97**, 823-29 (2005). This error has been corrected in the revised version (now as reference 31).

Reviewer #1 (Remarks to the Author):

The authors describe a novel crystal structure of a human aquaglyceroporin, AQP10. It is the first structure of a mammalian aquaglyceroporin, with the structure as well as functional data suggesting an intriguing pH gating. The findings are interesting and relevant not only for structural biologists and biophysicists, but also from a clinical point of view: AQP10's role in glycerol metabolism makes it a putative target in obesity treatment. The paper is well written and easy to follow, and the findings are relevant.

We thank the reviewer for the work and we are pleased to see that we share the opinion that our findings are highly interesting for a broad audience.

However, particularly the molecular dynamics simulation data are inconclusive and require extensive revision before I can recommend the paper for publication.

First, throughout the manuscript the pore radius is used as a proxy for permeation, with a radius of $> 0.8\text{\AA}$ presumed to be water permeable and a radius of $> \sim 1.2\text{\AA}$ presumed to be glycerol permeable. Why aren't permeabilities in MD used to validate this assumption?

In the manuscript we show that hAQP10 is permeable to water and glycerol (at low pH) *in vivo*, *in vitro* and based on structural analysis of the pore radii. Hence, the discussed permeation has already been extensively validated. Nevertheless, we agree with the reviewer that MD simulations can be used to shed light on water/glycerol permeabilities.

Is the crystallographic (closed) state indeed permeable to water, but not glycerol? Likewise, how is this different in the putatively open state? At least a glycerol permeation profile should be computed (and compared to e.g. GlpF) to be able to safely conclude that the putatively open state is indeed permeable to glycerol.

The crystal structure is likely water permeable as demonstrated by the performed HOLE analysis. In the simulations, following energy minimization and during equilibration, we observe a continuous water file along the pore of the determined hAQP10 structure, also suggesting that the 'closed' protein is water permeable. As outlined in the manuscript, we observe a widening of the pore (upon protonation of His80), and hence the open states also allow passage of water (however, not all loop configurations are equally open).

To address the glycerol permeability, we have revised our simulation procedure and performed new simulations. To get enhanced simulation speeds, we turned to graphical processing units (GPUs) and the ACEMD simulation engine. Thus, we are now able to base our conclusions on 4 simulations of 120 ns each, for both single (epsilon) and double protonated H80 (total of 8 simulations). Please, see also the revised "Materials and Methods" section.

Based on a PCA and cluster analysis of the combined trajectory, we obtain similar results as the previous simulations; the loop region for the double protonated system quickly changes into a more open configuration, while the majority of the single protonated simulations represent a configuration more resembling the crystal structure.

Nevertheless, we observe only a few glycerol permeation events for each protonation state occurring in various loop configurations, on a timescale that is in agreement with the one previously observed in computational studies of GlpF ($0.4\text{-}1\ \mu\text{s}^{-1}$; Lu et al. *Biophys J.* 2003;85(5):2977-87). However, as the permeation profile is originating from an ensemble of fast structurally interchanging states, it would require massive simulations in order to fully statistically quantify glycerol permeability. In this work we focus on the mechanistic aspects of gating and thus consider such simulations beyond the scope of the present study.

Second, the statistics of the observed opening remains unclear. Is the opening transition observed in all monomers, or only in some? Multiple transitions would be necessary for a statistically significant result.

Figs. 4a and Supplementary 8c and d show the combined trajectory of all monomers for both protonation states. From color coding of the determined clusters, it is evident that all double protonated monomers change configuration towards clusters 3, 4 and 5. Thus, transitions for all monomers are observed.

Third, has the pKa of H80 been measured or estimated, to test the hypothesis of a doubly protonated H80? And how would the pKa of the other H80 shift if the first has been doubly protonated? Presumably in the low dielectric of the membrane an interaction between the positive charges on the doubly protonated His might occur. Therefore, how probable is the state with all four H80 doubly protonated?

We have exploited PropKa to estimate pKa values, a method that considers the local environment, but not the membrane (Uranga et al. *Comput. Theor. Chem.* 2012;1000:75-84). PropKa computes pKa values of approximately 3.6 in monomers A, B and C of the crystal structure, while it is approximately 3.7 in monomer D. Even though the changes are minor, this shift suggests that monomer D is more prone to be double protonated, which is in agreement with the slightly more open pore in monomer D. Along that line, we have also determined the pKa of the representative structures of different clusters of the MD simulations presented in Fig. 4a. Here we also observe an increasing pKa upon opening of the pore, which supports the notion of His80 being the pH dependent switch in the hAQP10 gate region. Because of the large distance (about 35 Å to the most close) between the His80 of the different monomers, we assume that the His80 residues are triggered independently (the likelihood will depend on the internal pH).

Finally, how unique is the local environment around H80 for AQP10? Would it be feasible that other aquaporins or aquaglyceroporins share a similar mechanism?

As observed in the new Supplementary Fig. 5c, the local environment of H80 in hAQP10 is unique, with loop B adapting an entirely new configuration (not observed in previously determined structures of aquaglyceroporins) as allowed by the G73G74-motif which is not present in other human aquaglyceroporins. Thus, it is not likely that this mechanism is maintained in other aquaglyceroporins (or that it is modulated, as observed e.g. for the F85V mutation).

Reviewer #2 (Remarks to the Author):

Gotfryd et al. present a crystal structure of the human aquaglyceroporin 10 at 2.3 Å resolution. Besides the prokaryotic GlpF from *Escherichia coli* and the eukaryotic PfAQP from *Plasmodium falciparum* this is the third structure of a water/glycerol channel. As a special feature of hAQP10, the authors point out that the size exclusion selectivity filter (aromatic/arginine region) is considerably wider than in the other aquaporins and does not hold a glycerol molecule in the crystal structure. Further, their structure appears in a closed configuration with regard to glycerol towards the cytoplasmic end of the channel, whereas the smaller water might still pass. Due to the presence of a histidine in this area, a pH gating mechanism is proposed that opens the channel at low pH. The authors further try to corroborate this interpretation by functional measurements of wildtype and mutant hAQP10 in cell- and vesicle-based assays at different pH conditions.

There are, however, major inconsistencies here. The physiological context also does not fully fit into the scenario. Besides adipocytes where the pH may drop during lipolysis, AQP10 is prominently expressed in the small intestine and facilitates glycerol uptake at neutral pH. In adipocytes, why would pH-regulation of AQP10 be necessary (and kept during evolution) if a permanently open AQP10 would do the same trick and possibly even better (cattle obviously can dispense with AQP10 completely because it is a pseudo gene; Tanaka et al. *Biochem Biophys Res Commun* 2015 Mar 28;1:16-21).

We disagree that there are major inconsistencies in the data. Our findings are in line with the established phenomenon of cytoplasmic acidification triggered by lipolysis (see also our new data in the updated Fig. 1b and the accordingly revised “Materials and Methods” section), thereby allowing adipocytes to (increase) release of glycerol specifically under lipolytic conditions. In addition, the small intestine is characterized by an acidic microclimate near the brush border (Fallingborg *Dan Med Bull* 1999;46(3):183-96). This is, again, in agreement with our proposed pH-gating control of hAQP10. Furthermore, the expression profile of hAQP10 is not fully clear, complicating analysis of the role of hAQP10 in the small intestine; hAQP10 has been found in the apical membrane of intestinal villi (Mobasher et al., *Histochem Cell Biol* (2004) 121:463-471), in the capillary endothelium (small intestinal villi) and in the GEP endocrine cells (Li et al., *Biol. Cell* (2005) 97, 823-9). Finally, hAQP10 is not a pseudogene in humans, illustrating yet another difference between humans and cattle.

Specific points:

Suggested pH regulation:

1. In a reconstituted polymersome system, the authors show water and glycerol permeability of the human aquaglyceroporins AQP3, AQP7, AQP9, and AQP10 at neutral pH (Fig. 1c). When shifting to pH 5.5, water permeability appears unaffected but glycerol permeability of AQP3, AQP7, and AQP9 ceases, while it increases dramatically (about one order of magnitude) with AQP10. The data do not match with earlier data. AQP3 (Zelenina et al. *J Biol Chem* 2003 Aug 8;278(32):30037-43) have shown inhibition of AQP3 at low pH and identified involved residues at the extracellular face of the protein. Also recently AQP7 has been shown to be blocked by low pH from the extracellular side (Rothert et al. *J Biol Chem* 2017 Jun 2;292(22):9358-9364). However, in the same study, glycerol permeability of AQP9 was insensitive to pH changes, i.e. AQP9 was functional down to

pH 3. Therefore, AQP9 glycerol permeability should have been seen in the assay. Lack thereof renders the system and the data quite doubtful.

We do not agree that our data (presented in Fig. 1c) are doubtful. We observe similar results in proteopolymersomes and *in vivo*, and the findings agree with our structure. We have now also conducted pilot experiments employing a more classical proteoliposome-based system (please, see the plot below), where detergent-solubilized hAQP10 was reconstituted into liposomes (*E. coli*-derived lipids), and the obtained results that are in accordance with our previous findings from the proteopolymersome-based assay. As illustrated in the graph showing results from representative measurements, hAQP10 displays again increased glycerol permeability at low pH compared to physiological pH (green curve vs blue curve, respectively). Rate constants (k) for the glycerol conductance kinetic (obtained from single exponential fitting) were 0.47 and 0.38 sec⁻¹ for pH 5.1 and 7.4, respectively (0.18 sec⁻¹ for the control, i.e. empty, liposomes).

Legend: hAQP10 glycerol permeability functional assay. Normalized fluorescence intensity changes of hAQP10-*E. coli* lipid-derived proteoliposomes at pH 5.1 (blue) and pH 7.4 (green) compared with no-hAQP10 liposome control (black) in response to osmotic shock. Single exponential fitting was used to obtain average rate constants (k).

We are aware of the data on hAQP3 published by Zelenina et al. However, besides aquaporins, water permeability is strongly influenced by the membrane lipid composition, which contributes to the overall measured permeability. Zelenina's assay involved overexpression of hAQP3 in human BEAS-2b cells, i.e. a significantly more complex environment as compared with polymersomes (obviously lacking other integral membrane proteins that may influence permeability improving the sensitivity of our approach and a reason for why such an assay has been included in our work), which may explain the observed P_f difference for hAQP3. As for glycerol permeability, we show similar pH effect to reported by Zelenina et al. for water fluxes.

Concerning the findings published by Rothert et al. ("AQP9 was functional at all tested pH conditions, glycerol permeability decreased by 45% with increasing buffer acidity (Fig 1A, right))",

our data shown in our Fig. 1c are also in accordance: although at pH 5.5 P_{gly} was nearly not detected, glycerol permeability at pH 7.4 was markedly increased.

Thus, we conclude that qualitatively our results agree with those observed by de Almeida et al. (Mol Biosyst. 2016;12(5):1564-73), Zeuthen and Klaerke (J Biol Chem. 1999;274(31):21631-6) as well as Rothert et al (J Biol Chem. 2017 Jun 2;292(22):9358-64).

2. In the same figure (1b), vesicles prepared from adipocytes show only a mild, probably twofold increase in glycerol permeability at pH 5.5 with a weak statistic p-value of 0.037. Considering that AQP10 accounts for 50% of the glycerol flux at neutral pH (Laforenza et al. PLoS One. 2013;8(1):e54474), a much stronger effect should have been seen to match the data from the polymersomes. Again, how does this fit?

Silencing of hAQP10 in differentiated human adipocytes resulted in a 50 % decrease of glycerol permeability at neutral pH (Laforenza et al. PLoS One. 2013;8(1):e54474). In acidic conditions however, the overall glycerol permeability in primary adipocytes doubled and that represents the overall sum of glycerol flux mediated by all four aquaglyceroporins (i.e. hAQP3, 7, 9 and AQP10). Moreover, considering that glycerol permeation of hAQP3, 7 and 9 is reduced by acidic pH, the contribution of hAQP10 is likely more than two-fold. Furthermore, we have to highlight that results obtained in experiments performed on human tissue samples have in general greater variability. Hence, although the p-value reported in Fig. 1b is somewhat high, our findings are statistically significant.

3. pH-shifts in the yeast cell-based assays of AQP10 were done in the media, i.e. at the extracellular side, and not where the intracellular pH-gate is said to be located (or where the lipolysis-derived pH drop should occur in adipocytes). Still, a 4-6 fold increase in glycerol permeability is reported by the authors (Suppl. Table 5) not fitting their proposed gating mechanism.

It is correct that our yeast-cell assay was based on media controlling pH of the extracellular environment. However, the internal pH (pH_{in}) of the yeast cell system employed in the present work has already been shown to correlate with the external pH (Leitão et al. PLoS One. 2012;7(3):e33219) and this data were included in the original version of our manuscript (Fig. 3b, the lower x axis with pH as also indicated in the figure legend).

Yeast cells exposed to glucose depletion are not able to maintain the internal pH around 7 and change their cytosolic pH ultimately approaching extracellular pH (Orij et al. Microbiology. 2009 Jan;155(Pt 1):268-78; Dechant et al. EMBO J. 2010 Aug 4;29(15):2515-26.). In our experimental conditions, cells were glucose-starved for 45-60 min, the time taken to harvest the cells and load with fluorescent probe. Extracellular pH 7.4, 6.8 and 5 represent intracellular pH of 7.4, 6.8 and 6.1, respectively (Dechant et al. EMBO J. 2010 Aug 4;29(15):2515-26; Leitão et al. PLoS One. 2012;7(3):e33219).

Furthermore, when comparing yeast cells with adipocytes, another important phenomenon has to be considered. Lipid head groups have been shown to act as proton-collecting antennae, thereby leading to local acidification of the environment (Brändén et al. Proc Natl Acad Sci USA. 2006;103(52):19766-70). In adipocytes, the cytoplasm consists of a thin layer between plasma membrane and lipid droplet, hence always in close proximity of lipids. Thus, the microclimate near plasma membrane may become very acidic (especially upon stimulation of lipolysis). Therefore, the

observed *in vivo* effect of hAQP10-mediated glycerol export is most likely much more pronounced than that observed in the yeast-based assay (please, see also our response to point 2).

4. The AQP10 crystal structure was solved at pH 6.0, i.e. close to the pH of 5.5 at which the authors claim the channel to be open. However, they find a closed structure. Please explain.

Precise explanation of why a certain structure is obtained during crystallization is difficult. It is for example not uncommon to obtain different type of crystals (packing / space group) even within the same crystallization drop (experiment). Even the cryo-cooling used to preserve crystal samples can introduce some degree of the structural bias (Fraser et al. Proc Natl Acad Sci USA. 2011 Sep 27;108(39):16247-52).

One reason for why the structure is closed may be how the pH is assessed. Typically, the “Materials and Methods” section of structural biology manuscripts (including this work) lists the pH of the buffer used in the reservoir solution (6.0 in our case). The final pH depends, however, also on other components (e.g. salts, precipitants, additives, detergents and/or the protein itself). Furthermore, additional pH changes may take in the mother liquor (i.e. after crystals are formed).

It is also possible that crystal packing, which is not determined exclusively by pH, but by all components (again salt, precipitant, detergent, additives etc.), affects the structure. Our analysis of the crystal packing (shown in Supplementary Fig. 7) suggests however little influence of the crystal contacts on the structure (a maintained AQP fold is observed, few interacting molecules or crystal contacts are present close to the classical selectivity filter and a monomer specific pattern binding pattern of detergents renders effects on the novel gate region unlikely).

Finally, we wish to note that the structure and our interpretations thereof are validated through *in vitro* and *in vivo* characterization and through MD simulations and consequently we are convinced that they are correct.

5. Ishii et al. (Cell Physiol Biochem. 2011;27(6):749-56) even suggested a dual functional mechanism of AQP10 that requires trans-stimulation by smaller concentrations of glycerol at the lower end of the transmembrane gradient. This has not been considered in the assays or stimulations.

We are familiar with the findings published by Ishii et al. Despite decades of work with an arsenal of different techniques, very few reports suggest that aquaporins function as carriers, as proposed in the mentioned paper. We also do not observe indications of such a behavior in any of our data (crystal structure, functional characterization *in vivo* and *in vitro* or MD simulations) and we are not aware of any other report on hAQP10 that even hints at such a mechanism. We cannot exclude that such regulation is present, but the structure strongly points towards pH-gating through the H80 region and so do our complementing data.

Wide ar/R region:

5. The feature of a particularly wide ar/R filter is not interpreted in a physiological context. Only a brief statement questioning “if the functional role of the ar/R region is maintained in hAQP10” is given.

The text has now been revised and the findings put in context. The manuscript now reads: “These observations indicate that the functional role of the ar/R region may not be maintained in hAQP10 (the ar/R filter of orthodox aquaporins is typically very tight, excluding glycerol passage, whereas in aquaglyceroporins it also permits flow of certain small solutes, but prevents most other compounds (Supplementary Fig. 1)” and includes an additional reference (Finn and Cerdà Biol Bull. 2015;229(1):6-23).

Reviewer #3 (Remarks to the Author):

In this manuscript, Gotfryd et al. present the structural and functional studies on human aquaglyceroporin AQP10 to provide the molecular basis of its pH-dependent regulation. The authors first show that glycerol flux into human adipocytes is pH-dependent, and pinpoint AQP10 as a responsible protein for such pH-dependent glycerol flux. The authors next describe the structure of human AQP10, determined at 2.3 Å resolution, which reveals the unique features of AQP10 as compared to other AQPs: 1) the wide ar/R selectivity filter, 2) the cytoplasmic restriction formed by F85, and 3) the loop B capping the cytoplasmic vestibule. Combined with structural, biochemical and computational analyses, the authors propose a pH-gated glycerol permeation mechanism of AQP10 where His80 and Phe85 play an important role.

The major achievement here is the structure determination of a human aquaglyceroporin; it is for the first time the structure of a mammalian aquaglyceroporin is presented. Although the structures of bacterial and parasite aquaglyceroporins have already been reported, the present structure will allow a more systematic comparison of aquaglyceroporins across different organisms. The structure also provides an unexpected finding; the loop B forms a constriction at the cytoplasmic vestibule, which is likely to be responsible for pH-gating. Although the data presented here are not sufficient to derive a strong conclusion on how pH-gating is achieved through this loop, the high-resolution structure nevertheless provides a basis for further functional and computational analyses on this aspect.

Overall, this is an important study that represents a significant advancement in understanding glycerol permeation in human cells through AQPs. I recommend this manuscript for publication in Nature Communications, pending some revisions as detailed below.

We thank the reviewer for the positive evaluation, constructive criticism and for recommending our work to be published in Nature Communications.

1) A major drawback of this manuscript is the lack of clear, logical explanation on how low pH, at which AQP10 opens, is related to fat metabolism in the physiological context. While the pH-gating mechanism of AQP10 itself is plausible, how such pH-dependent glycerol permeation impacts human fat diet is poorly discussed. Please provide more physiological insights from other literature, if any.

Our manuscript reveals a correlation between pH-dependent hAQP10 activation at low pH, with acidification of adipocytes observed during lipolysis. To further bridge even physiological events, we have now obtained data showing a decrease of intracellular pH and elevation of glycerol release using human primary adipocytes under lipolytic condition (isoproterenol exposure), in contrast to insulin-treated (lipogenesis induction). These results are incorporated to the revised Fig. 1b (pH

measurements) and as a new Supplementary Fig. 2 (glycerol release), respectively. Naturally, the “Materials and Methods” section has been updated accordingly. Moreover, we have updated a sentence from the discussion section to pinpoint the possible physiological role of hAQP10 in the duodenum. The sentence now reads: “Glycerol flux across plasma membranes of adipocytes (and likely duodenal enterocytes, where reported pH was shown to be acidic) is demonstrated to be stimulated by low pH...” and includes an additional reference (Fallingborg *Dan Med Bull.* 1999 Jun;46(3):183-96).

2) From the manuscript it is not clear whether the structure is of an open (glycerol-conductive) or a closed state. I suggest adding a more clear statement about the conductive state of the channel. The authors argue that the protonation of His80 is the central mechanism of the gating at cytoplasmic site, in the latter part of this paper. The reviewer thinks this is very interesting, and also thinks confirming the protonation state of the His80 in this crystal structure is important. The pKa of the histidine residue is 6.04, so in this crystallizing condition, it is possible that His80 is double protonated. In that case, the double protonated His80 will also stabilize the closed state. So the reviewer suggest that the authors should calculate the pKa of His80 and discuss about the protonation state of His80.

As outlined in the manuscript, the obtained crystal structure is likely permeable to water, but not glycerol; see for example “Notably, HOLE analysis suggests that this narrowing (0.9 Å) permits water (0.8 Å at the ar/R filter in hAQP2), but not glycerol (1.3 Å in AqpM) flux”. We have exploited PropKa to estimate pKa values, a method that takes into account the local environment, but not the membrane (Uranga et al. *Comput. Theor. Chem.* 2012;1000:75-84). PropKa predicts pKa values of approximately 3.6 in monomers A, B and C of the crystal structure, while it is approximately 3.7 in monomer D. Even though the changes are minor, this shift suggests that monomer D is more prone to be double protonated, which is in agreement with the slightly more open pore in monomer D. Along that line, we have also determined the pKa of the representative structures of different clusters of the MD simulations presented in Fig. 4a. Here we also observe an increasing pKa upon opening of the pore, which supports the notion of His80 being the pH dependent switch in the hAQP10 gate region.

3) The omit electron density map in Supplementary Figure 3 shows that glycerol molecule is only partially visible. This raises question as to whether the observed density really belongs to a glycerol, or other molecules such as partially ordered water. Have the authors tried refinement by placing water molecules instead of a glycerol? Related to this, it is not clear whether this glycerol-binding site correspond to the previously observed glycerol-binding sites in other AQPs? I suggest adding a figure comparing the glycerol-binding sites of different AQPs.

We thank the reviewer for these suggestions. As shown in the new Supplementary Fig. 4c, we have also attempted to place two water molecules instead of the glycerol molecule, However, also in the presence of these two waters, model is missing in the region (unassigned Fo-Fc density), supporting the notion that the density belongs to a glycerol molecule. As suggested by the reviewer, we now display in Supplementary Fig. 5a an overlay of glycerol and water molecules in other aquaglyceroporins. They all have two or more glycerol molecules present in the pore, while hAQP10 only has a single glycerol molecule. The arrows in this figure correspond to water and glycerol molecules related to hAQP10 (gray spheres).

4) Most structural description in the manuscript focus the residues on loop B, but they are poorly represented in the figures. I suggest including figures depicting the structural elements below.

- The position of the G73G74-motif.
- The distances between E27, H80, and R94 to show whether these residues are within the hydrogen-bonding distances. How F85 and R94 stabilize the loop B should be referred.
- The distance between F85 and glycerol.
- Other possible hydrogen-bonding or hydrophobic interactions in this region.
- Structural comparison of the loop B from AQP10 and different aquaporins.
- How the loop B interacts with the cytoplasmic gate is not shown in the Fig. 2-c. The close view of the loop B will be needed to discuss the “tight arrangement”.

Supplementary Fig. 5 has now been revised with a variety of views of loop B region. The GG-motif has been highlighted by an arrow in all relevant panels. Relevant distances (mentioned by the reviewer) are displayed in panel d. An overlay of relevant aquaglyceroporins is shown in panel c demonstrating the uniqueness of the loop B in hAQP10. The tight arrangement of the hAQP10 gate is shown in both panels e and f.

5) The authors predict that the pH-sensitivity is missing in other human aquaglyceroporins because of the valine replacement of F85. The authors could test this possibility by measuring the pH-dependent glycerol flux of F85V mutant.

We have now tested the suggested mutant in our yeast-cell based assay. Similarly to G73A mutation, F85V mutant displays vast increase in the glycerol flux at low pH. We were not able to precisely measure P_{gly} at pH 5.1 (the rough estimation is at least 30) as well as P_f (due to a very rapid glycerol entry). Hence, in the Fig. 3c and Supplementary Fig. 6e results obtained for this mutant are not shown, but they are mentioned in the text and legends. However, in line with our predictions, the function is affected and the observed glycerol specific pH dependence in the wild-type form is lost, in line with the H80 region being critical for the function. We also note that a pH dependency remains, but the local environment in the other human aquaglyceroporins is likely also affected by the missing GG kink that places loop B in the tight configuration observed in the hAQP10 structure.

6) The authors argue that “All-in-all, based on structural, functional and MD simulation analyses, we propose a pH-dependent gating mechanism of hAQP10 triggered by protonation of H80, which at low pH reorients (from chain A, D and cluster #1), stabilized by E27 (clusters #2-4) (Fig. 4a).”. In this argument, the electrostatic interaction between H80 and E27 caused by the protonation of H80. This is the major topic of discussion, so the substitution mutation analysis such as E27Q is essential.

We have now tested the suggested mutant in our yeast-cell based assay (see revised Fig. 3c and Supplementary Fig. 6e). As expected, the pH dependency of this mutant is abolished.

7) The result and the discussion of the MD simulation seems to be consistent with those in the paper “ref 23 (S. Kaptan et al., H95 Is a pH-Dependent Gate in Aquaporin 4. Structure 23, 2309-2318(2015))”. The reviewer thinks that the paper “ref 23” should also be discussed in the discussion section.

We already referred to this particular reference in the original version of the manuscript and prefer to refrain from the proposed discussion for clarity.

8) I would like to know how the proposed pH-gating mechanism of AQP10 compares to that proposed for other AQPs.

The pH-gating in hAQP4 has been proposed to be triggered by the equivalent residue to H80 in hAQP10 (which is a histidine also in hAQP4). The available structural information of hAQP4 suggests an open channel (in contrast to the structure presented in this work). Furthermore, the structures are rather different in the gate region (hAQP4 lacks the GG motif that allows the unique arrangement of loop B in hAQP10), suggesting that the mechanisms are only linked remotely (shared histidine pH trigger, but likely the details of the gating are rather different). Noteworthy, His or Ser protonation-dependent mechanisms were also proposed for plant PIP2 aquaporins (Tournaire-Roux et al. *Nature*. 2003;425:393-7; Yaneff et al. *Biochim Biophys Acta*. 2016;1858:2778-87).

9) The reviewer thinks that the distribution of the distance between H80 and E27 in the MD simulation is very important, so these data (Supplementary Fig. 8a-b) should be shown in main Fig. 4.

We agree with the reviewer that the distribution of distance between H80 and E27 is important, but we prefer to leave this information in the Supplementary Material for clarity.

Reviewers' Comments:

Reviewer #1:

Remarks to the Author:

The authors have addressed most of my concerns in the revised manuscript. However, one critical point remains: it is not convincingly demonstrated that the putative "open" state is indeed glycerol permeable. As such, the manuscript describes an interesting hypothesis and a plausible model rather than demonstrating an AQP10 gating mechanism. I don't think it's valid to argue, as the authors do, that the study of glycerol permeability is "beyond the scope of the present study" as it is exactly what would be required to justify the conclusions of the present study.

In addition, the included propka analysis raises additional questions. First, the reported pKa values of 3.6-3.6 do not explain a double protonated His80 at the measured pH of >5? This should be discussed. Second, the minute difference between the calculated pKa of monomers A,B,C (3.6) and D (3.7) does not appear to be statistically significant, unless the propka accuracy is $\ll 0.1$ pKa units, which I do not believe is the case.

Reviewer #2:

Remarks to the Author:

The authors have addressed the raised issues well. However, the impression remains that the physiological relevance of AQP10 pH-gating is overemphasized. The authors show in Fig. 1c a decrease of the intracellular pH by about one tenth of a pH unit (which is in accordance with earlier studies, see Rudman, D. & Shank, P. W. 1966, *Endocrinology*. 79, 1100ff, and Civelek, V. N. et al. 1996, *Proc Natl Acad Sci U S A*. 93, 10139ff). It is questionable (and has not been shown by the authors) if such small physiological variations in intracellular pH will in fact trigger relevant changes in AQP10 glycerol permeability.

Together, the ms shows a new aquaglyceroporin structure, which is fine and interesting; statements regarding the physiological relevance should be phrased much more carefully throughout the ms (e.g. second sentence of the abstract!).

Reviewer #3:

Remarks to the Author:

This author is satisfied with this revision and now recommend the publication in *Nature Communications*.

Point-to-point response to the Reviewers' comments

We thank the editor for the swift handling of our paper, and the three reviewers for again recognizing the merit of our work. As detailed below, we have addressed the Reviewers' comments fully and carefully, and revised the manuscript accordingly. To facilitate the reviewing process, we have copied the Reviewers' original comments, which are shown in **black**. Our responses are then shown in **blue**.

Reviewer #1 (Remarks to the Author):

The authors have addressed most of my concerns in the revised manuscript. However, one critical point remains: it is not convincingly demonstrated that the putative "open" state is indeed glycerol permeable. As such, the manuscript describes an interesting hypothesis and a plausible model rather than demonstrating an AQP10 gating mechanism. I don't think it's valid to argue, as the authors do, that the study of glycerol permeability is "beyond the scope of the present study" as it is exactly what would be required to justify the conclusions of the present study.

In addition, the included propka analysis raises additional questions. First, the reported pKa values of 3.6-3.6 do not explain a double protonated His80 at the measured pH of >5? This should be discussed. Second, the minute difference between the calculated pKa of monomers A,B,C (3.6) and D (3.7) does not appear to be statistically significant, unless the propka accuracy is $\ll 0.1$ pKa units, which I do not believe is the case.

In the previous point-to-point response (page 3), we already reported that we observe glycerol permeation in the MD simulations. For clarity, data for simulations with double protonated H80 (e.g. the open structure) is now included as a new figure (Supplementary Fig. 10), clearly indicating passage of glycerol, and addressing the final remaining serious concern raised by the reviewer.

The estimated His80 pKa values range from 3.6 (for closed structure) to 7.1 (for the most open, last cluster in Fig. 4), suggest that the generated crystal structure and the observed clusters in the MD simulations display different His80 protonation susceptibility (at relevant intracellular pH, e.g. pH \sim 6, the double protonation probability is low for the closed structure, while the double protonation probability is high for the open clusters). This is in complete agreement with the notion of allowed glycerol conductance at low(er) pH.

We agree with the reviewer that the estimated pKa values may not necessarily be significant, and we have therefore revised the main text and Fig. 4 accordingly. The important trend is however that mono protonation of His80 (at intracellular pH of \sim 6, above the estimated pKa) is associated with closed structures (crystal structure & early clusters in Fig. 4), while double protonation of His80 (at intracellular pH of \sim 6, below the estimated pKa) is associated with open structures (late clusters in Fig. 4).

Reviewer #2 (Remarks to the Author):

The authors have addressed the raised issues well. However, the impression remains that the physiological relevance of AQP10 pH-gating is overemphasized. The authors show in Fig. 1c a decrease of the intracellular pH by about one tenth of a pH unit (which is in accordance with earlier studies, see Rudman, D. & Shank, P. W. 1966, *Endocrinology*. 79, 1100ff, and Civelek, V. N. et al. 1996, *Proc Natl Acad Sci U S A*. 93, 10139ff). It is questionable (and has not been shown by the authors) if such small physiological variations in intracellular pH will in fact trigger relevant changes in AQP10 glycerol permeability.

Together, the ms shows a new aquaglyceroporin structure, which is fine and interesting; statements regarding the physiological relevance should be phrased much more carefully throughout the ms (e.g. second sentence of the abstract!).

According to reviewer's request, we have softened statements regarding the physiological relevance throughout the manuscript (including the abstract).

Reviewer #3 (Remarks to the Author):

This author is satisfied with this revision and now recommend the publication in Nature Communications.

We have no further comments/amendments.

Reviewers' Comments:

Reviewer #1:

Remarks to the Author:

The authors have addressed most of my concerns in the revised manuscript, and I appreciate the inclusion of glycerol permeation data from the MD simulations. However, the shown permeation pathway in the new figure S10 appears odd, as if the glycerol approaches the residues H80, R94 and S77 on the permeation pathway with zero distance, i.e. would actually clash with them?

The pKa concerns have been satisfactorily addressed.

Point-to-point response to the Reviewers' comments

We thank the editor for the swift handling of our paper, and the three reviewers for again recognizing the merit of our work. As detailed below, we have addressed the final remark from Reviewer #1 fully and carefully, and revised the manuscript accordingly. To facilitate the reviewing process, we have copied the Reviewer's remaining comments, which are shown in **black**. Our responses are then shown in **blue**.

Reviewer #1 (Remarks to the Author):

The authors have addressed most of my concerns in the revised manuscript, and I appreciate the inclusion of glycerol permeation data from the MD simulations. However, the shown permeation pathway in the new figure S10 appears odd, as if the glycerol approaches the residues H80, R94 and S77 on the permeation pathway with zero distance, i.e. would actually clash with them?

The pKa concerns have been satisfactorily addressed.

Before we can make a decision, I would kindly ask you to comment on the last concern (the clash of glycerol with some residues). Could you please send me your explanation via email.

We have conducted the refinement of the crystal structure with standard restraints that prevent clashes (indeed there are no major clashes in the crystal structure, as indicated by the previously supplied validation report).

Furthermore, the MD simulations were run with the CHARMM22 and CHARMM36 parameter sets for the protein and the glycerol, respectively. These force fields include both bonded and non-bonded parameters for all atoms. The non-bonded interactions include a Lennard-Jones potential that ensures repulsion of two atoms at short distances. Thus, if the distance between glycerol and protein was approaching zero, it would result in infinite repulsion energy; consequently, in a simulation crash that was not observed for any of our simulations.

Hence, the remaining question must rather relate to a visualization issue. The glycerol is not clashing with the protein residues; however, this may not be completely clear from the previous version of Fig. S10. We therefore attach clarifying panels to Fig. S10 with the very same MD simulation snapshots as shown in the previous version of Fig. S10, but with each MD simulation snapshot (or crystal structure) shown in two different views. The closest distance from the glycerol molecule to any of the mentioned residues is also shown, clearly demonstrating that no clashes between glycerol and protein occur.

We hope that this explains the final question raised by Reviewer #1 and that a rapid positive decision will be possible, considering that there is an unreleased structure in the PDB of a related protein as mentioned in the previous cover letter.